# Full assembly of HIV-1 particles requires assistance of the membrane curvature factor IRSp53

**Kaushik Inamdar[1], Feng-Ching Tsai[2], Rayane Dibsy[1], Aurore de Poret[1], John Manzi[2], Peggy Merida[1], Remi Muller[3], Pekka Lappalainen[4], Philippe Roingeard[5], Johnson Mak[6], Patricia Bassereau[2], Cyril Favard[1†], Delphine Muriaux[1]\***

[1]Infectious disease Research Institute of Montpellier (IRIM), CNRS UMR 9004, University of Montpellier, Montpellier, France; [2]Institut Curie, Université PSL, Sorbonne Université, CNRS UMR168, Laboratoire Physico Chimie Curie, Paris, France; [3]CEMIPAI, CNRS UAR3725, University of Montpellier, Montpellier, France; [4]Institute of Biotechnology, University of Helsinki, Helsinki, Finland; [5]MAVIVH UMR Inserm U1259, University of Tours, Tours, France; [6]Institute for Glycomics, Griffith University, Brisbane, Australia

**Abstract** During HIV-1 particle formation, the requisite plasma membrane curvature is thought to be solely driven by the retroviral Gag protein. Here, we reveal that the cellular I-BAR protein IRSp53 is required for the progression of HIV-1 membrane curvature to complete particle assembly. siRNA-mediated knockdown of IRSp53 gene expression induces a decrease in viral particle production and a viral bud arrest at half completion. Single-molecule localization microscopy at the cell plasma membrane shows a preferential localization of IRSp53 around HIV-1 Gag assembly sites. In addition, we observe the presence of IRSp53 in purified HIV-1 particles. Finally, HIV-1 Gag protein preferentially localizes to curved membranes induced by IRSp53 I-BAR domain on giant unilamellar vesicles. Overall, our data reveal a strong interplay between IRSp53 I-BAR and Gag at membranes during virus assembly. This highlights IRSp53 as a crucial host factor in HIV-1 membrane curvature and its requirement for full HIV-1 particle assembly.

**\*For correspondence:**
delphine.muriaux@irim.cnrs.fr

[†]ORCID: 0000-0002-8304-2980

## Introduction

The cell plasma membrane is a dynamic structure, where crucial processes such as endocytosis and exocytosis take place through local membrane deformations. Several pathogens, such as bacteria and enveloped viruses, interplay with the plasma membrane in the course of their replication cycle. Pathogens often enter the cells by endocytosis (*Grove and Marsh, 2011*; *Gruenberg and van der Goot, 2006*) and exit by membrane vesiculation (*Rheinemann and Sundquist, 2020*; *Welsch et al., 2007*), which are processes linked to the generation of plasma membrane curvature; either inward or outward. HIV-1 is an enveloped positive-strand RNA virus belonging to the family *Retroviridae*, and it is known to assemble and bud outward from the host cell plasma membrane (*Coffin et al., 1997*). The structural Gag polyprotein of HIV-1, by itself, is responsible for particle assembly (*Gheysen et al., 1989*), and it can oligomerize at the inner leaflet of the plasma membrane forming virus-like particles (VLPs). The force required to bend the membrane to achieve VLP formation has been proposed to be provided by Gag self-assembly (*Hurley et al., 2010*). The self-assembly of Gag has also been recently shown to segregate specific lipids (*Favard et al., 2019*; *Yandrapalli et al., 2016*) and proteins (*Sengupta et al., 2019*), generating plasma membrane domains that could favor budding (*Foret, 2014*; *Lipowsky, 1993*). However, only a small proportion

of Gag-initiated clusters reach the fully assembled state leading to VLP release in living CD4$^+$ T cells (*Floderer et al., 2018*), while a majority of these clusters lead to aborted events. Therefore, the mechanism by which the virus overcomes the energy barrier associated with the formation of the full viral bud remains an open question. Recently, a coarse-grained model of HIV assembly has shown that the self-assembly of Gag might not be sufficient to overcome this energy barrier (*Pak et al., 2017*), leaving the assembly in intermediate states. This supports the fact that other factors may be necessary to assist Gag self-assembly during the generation of new VLPs.

Plasma membrane curvature can also be generated by diverse host cell proteins. For example, I-BAR domain proteins sense and induce negative membrane curvature at the nanometer scale (a few tens to one hundred nanometers), that is, in the HIV-1 particle diameter size range, while generating outward micrometer-scale membrane protrusions such as membrane ruffles, lamellipodia, and filopodia. IRSp53 was first discovered as a substrate phosphorylated downstream of the insulin receptor (*Yeh et al., 1996*). It is also the founding member of the membrane curving I-BAR domain protein family, whose other mammalian members are MIM (missing-in-metastasis), ABBA (actin-bundling protein with BAIAP2 homology), PinkBAR (planar intestinal and kidney-specific BAR domain protein), and IRTKS (insulin receptor tyrosine kinase substrate) (*Zhao et al., 2011*). The latter, IRTKS, displays functional redundancy with IRSp53 (*Chou et al., 2017*; *Millard et al., 2007*) in being able to curve membranes. In addition to interactions with the plasma membrane, IRSp53 binds both Rac1 through its N-terminal I-BAR domain (*Miki et al., 2000*) and Cdc42 directly through its unconventional CRIB domain (*Krugmann et al., 2001*), and also downstream effectors of these GTPases such as WAVE2, Mena, Eps8, and mDia can bind IRSp53 through the SH3 domain. Thus, IRSp53 functions as a scaffold protein for the Rac1/Cdc42 cascade (*Scita et al., 2008*). IRSp53 was reported to exhibit a closed inactive conformation that opens synergistically upon binding to Rac1/Cdc42 and effector proteins (*Disanza et al., 2013*; *Kast et al., 2014*; *Miki and Takenawa, 2002*; *Suetsugu et al., 2006*). Regulation of IRSp53 activity was recently shown to occur through its phosphorylation and interaction with 14-3-3 (*Kast and Dominguez, 2019*). Structurally, the I-BAR domain of IRSp53 is composed of a rigid six alpha-helix bundle dimer that is *crescent*-shaped. Due to its concave membrane-binding surface and lipid interactions, IRSp53 is able to generate negative membrane curvature (*Zhao et al., 2011*). While capable of forming homo-dimers, IRSp53 is also able to recruit and form hetero-dimers with other proteins to form clusters for the initiation of membrane curvature (*Disanza et al., 2013*).

Since the Rac1/IRSp53/Wave2/Arp2/3 signaling pathway is involved in the release of HIV-1 particles (*Thomas et al., 2015*), we hypothesized that IRSp53 may be a prime candidate for membrane remodeling required during viral bud formation. Hence, we investigated the possible role of IRSp53 and its membrane curvature generating activity in HIV-1 Gag assembly and particle budding. Importantly, we discovered that IRSp53 is present in an intracellular complex with HIV-1 Gag at the cell membrane, incorporated in Gag-VLPs and it is associated with purified HIV-1 particles, supporting IRSp53's function in HIV-1 assembly as a facilitator of optimal HIV-1 particle formation through its membrane-bending activity. Thus, we identified IRSp53 as an essential non-redundant novel factor in HIV-1 replication, and demonstrated that it is critical for efficient HIV-1 membrane curvature and full assembly at the cell plasma membrane.

## Results

### IRSp53 knockdown decreases HIV-1 Gag particle release by arresting its assembly at the cell plasma membrane

We report here that the partial knockdown of IRSp53 expression reduces HIV-1 particle release in host Jurkat T cells and in the model cell line HEK293T (*Figure 1a,b*), similar to our previously reported results in primary T lymphocytes (*Thomas et al., 2015*). Cells were treated with siRNA targeting IRSp53 or IRTKS (validated by extinction of the transfected ectopic IRSp53-GFP or IRTKS-GFP proteins – *Figure 1—figure supplement 1b and c*, respectively). In Jurkat T cells, we expressed the viral Gag proteins in the context of HIV-1(ΔEnv) in order to only monitor the late steps of the viral life cycle. Partial IRSp53 gene expression knockdown (resulting in a maximum of 50% protein depletion) reduced particle release threefold as compared to the control siRNA (*Figure 1a*, bottom), and a sixfold reduction was determined when taking into account the percentage of protein depletion.

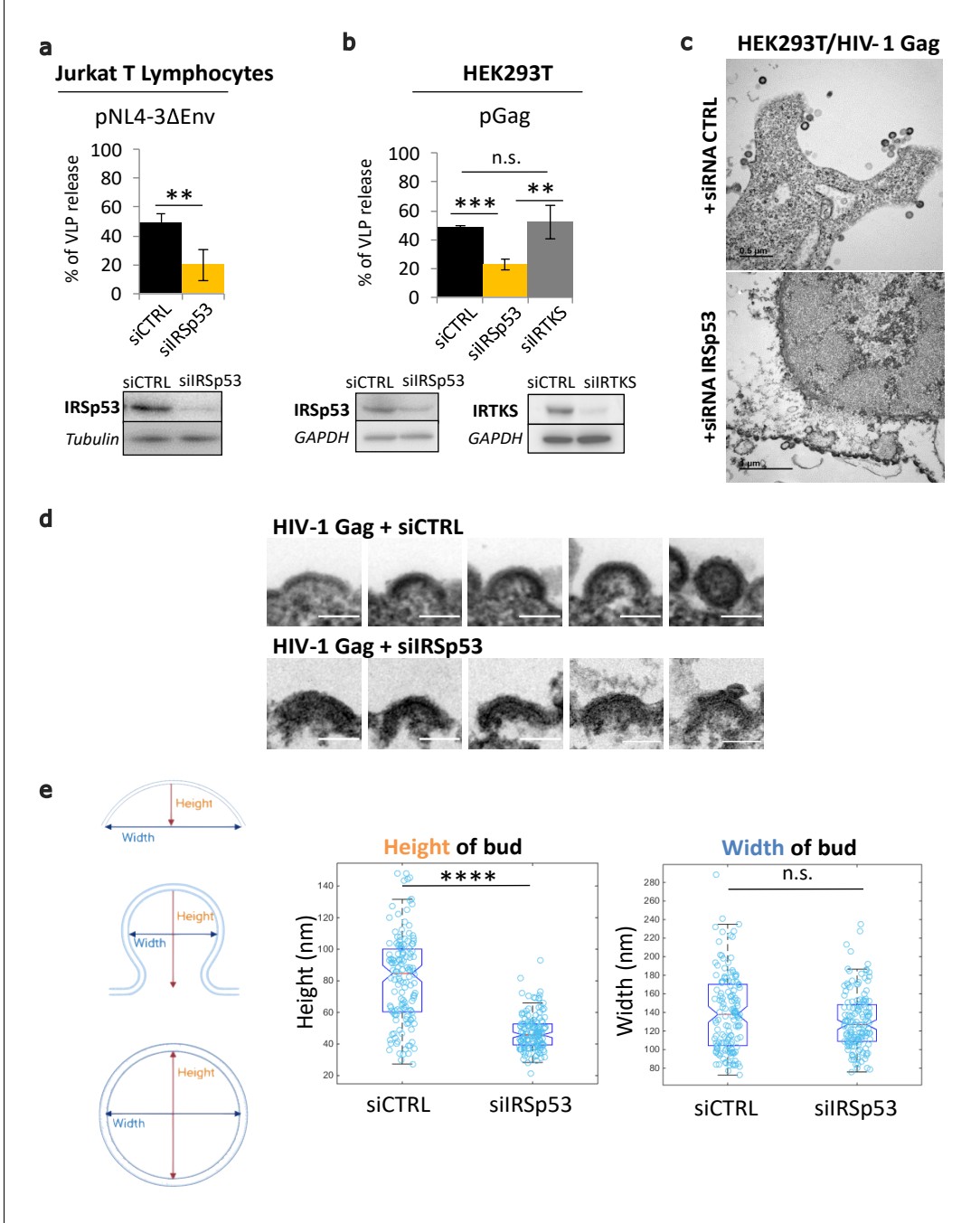

**Figure 1.** Partial knockdown of IRSp53 decreases HIV-1 Gag particle release by arresting assembly at the cell plasma membrane. (a) siRNA knockdown of IRSp53 expression in Jurkat T lymphocytes leads to a significant decrease in pNL4-3ΔEnv Gag particle release (see graph and immunoblots for IRSp53, $p = 0.00265$, Student's $t$-test, and loading controls beneath the graph). (b) Similarly, knockdown of IRSp53 expression in Gag expressing HEK293T cells led to a significant decrease in HIV-1 Gag particle release ($p = 0.00487$, Student's $t$-test), as compared to siRNA IRTKS ($p = 0.0116$, Student's $t$-test). On the other hand, knockdown of IRTKS expression (a closely related I-BAR protein) did not have a significant effect on particle release ($p = 0.0924$, Student's $t$-test, upper graph, immunoblots for IRSp53, IRTKS, and loading controls beneath the graph) ($n = 3$ independent experiments). Another multiple comparisons statistical test ANOVA was applied to compare the three siRNA conditions showing a significative difference with a p value = 0.0089. (c) Transmission electron microscopy images of HEK293T cells expressing HIV-1 Gag with siRNA control (upper panel) and siRNA IRSp53 (lower panel). Scale bar is 0.5 µm (upper image) and is 1 µm (lower image). (d) Transmission electron microscopy zoomed images of viral buds from HIV-1 Gag expressing cells treated with siRNA-mediated knocked down of IRSp53 (lower panel) showing arrested buds at the plasma membrane as compared to the siRNA control cells (upper panel) which display a normal range of buds in different stages of assembly and budding (scale bar = 100 nm). (e) Measurement of the bud dimensions (height and width median with interquartile) in the control siRNA and siRNA

*Figure 1 continued on next page*

*Figure 1 continued*

IRSp53 conditions (*n* = 145 buds from 14 different cells for each condition, *n* = 2 independent experiments). The knocked down cells exhibit a narrow range of heights corresponding to the arrested buds visible in the images, while the control cells display a wider range of heights corresponding to assembly progression (left graph, 'Height of bud'). Distribution of the height values in the two conditions is significantly different ($p = 1.05 \times 10^{-28}$, Kolmogorov-Smirnov test). On the opposite, the widths of the buds in both conditions did not display significant differences in distributions ($p = 0.0609$, Kolmogorov-Smirnov test).

The online version of this article includes the following source data and figure supplement(s) for figure 1:

**Source data 1.** Immunoblot Quantification using Fiji for *Figure 1B* graph.
**Source data 2.** Heights and Widths of HIV-1 Gag bud measurements for *Figure 1E* graph.
**Figure supplement 1.** Effect of siRNA-based knockdown of IRSp53 and IRTKS gene expression on HIV-1 Gag particle release.
**Figure supplement 2.** Protein sequence comparison of IRSp53 and IRTKS.
**Figure supplement 3.** Transmission electron microscopy of siRNA treated HEK293T cells expressing HIV-1 Gag.

This reduction in HIV-1 particle release is highly significant (*n* = 3 independent experiments, *p* value = 0.00265, Student's *t*-test, and ANOVA statistical test for multiple comparisons indicate *p* value = 0.0089) since the gene editing of IRSp53 cannot be complete, nor edited by CRISPR/Cas9 knockout, without being toxic for the cells. To compare the role of different I-BAR domain proteins from the same family, we also measured the effect of siRNA targeting IRSp53 and IRTKS (*Figure 1b*) on HIV-1 Gag VLP production in HEK293T cells (*Figure 1—figure supplement 1a*, see graph *Figure 1b*, and immunoblots *Figure 1—figure supplement 1d,e*). IRTKS shares similar protein domain organization and high sequence homology with IRSp53 (40% amino acid sequence identity and 59% sequence similarity, *Figure 1—figure supplement2*), and displays some functional redundancy with IRSp53 (*Chou et al., 2017*). IRTKS can also induce plasma membrane curvature (*Saarikangas et al., 2009*). Partial knockdown of IRSp53 (~50% protein depletion) resulted in a two- to threefold decrease in HIV-1 Gag particle production (*n* = 3 independent experiments, *p* value = 0.000487, Student's *t*-test) (*Figure 1b*, bottom, *Figure 1—figure supplement 1d*). In contrast, knockdown of IRTKS (*Figure 1b*, *Figure 1—figure supplement 1e*) did not have any significant effect on HIV Gag particle release (*n* = 3 independent experiments, *p* value = 0.0924, Student's *t*-test), thus precluding the possibility of redundant functions between IRSp53 and IRTKS in the context of HIV-1 Gag particle formation.

Electron microscopy imaging of siRNA IRSp53 treated HEK293T cells expressing HIV-1 Gag revealed arrested particle budding at the cell plasma membrane (*Figure 1c*, lower panel), as compared to the siRNA-control cells (*Figure 1c*, upper panel). While the control cells exhibited the normal phenotype of Gag-VLP budding from the cell plasma membrane, the IRSp53 knockdown cells displayed a series of viral buds arrested in assembly decorating the cell plasma membrane (*Figure 1c*, *Figure 1—figure supplement 3*). These results revealed an arrest in Gag assembly at the membrane and thus the involvement of IRSp53 in the assembly process. Since IRSp53 is an I-BAR protein involved in cell membrane curvature, we measured the curvature exhibited by HIV-1 buds in IRSp53 knockdown cells. While control cells displayed a range of HIV-1 Gag particles at different stages of assembly and budding, the cells knocked down for IRSp53 instead displayed arrested buds at an early assembly stage (*Figure 1d*). By measuring the dimensions of these arrested buds, we found that buds from cells knocked down for IRSp53 displayed a narrower range of curvature height (48 ± 22 nm), as compared to the control (85 ± 53 nm) (*n* = 145 buds from 14 different cells, *p* value = $1.053 \times 10^{-28}$, Kolmogorov-Smirnov test), while the bud widths presented no difference between siIRSp53 (135 ± 64 nm) and the control (140 ± 87 nm) (*n* = 145 buds from 14 different cells, *p* value = 0.0609, Kolmogorov-Smirnov test) (*Figure 1e*). The control cells thus exhibited a range of heights and widths consistent with the range of buds seen at the membrane of these cells. The result indicates that in the absence of IRSp53, the viral buds were unable to progress beyond a certain curvature.

## HIV-1 Gag expression in cells increases IRSp53 membrane binding and allows their complexation

Since both Gag and IRSp53 target the cell plasma membrane upon interaction with PI(4,5)P$_2$ (*Favard et al., 2019*; *Mattila et al., 2007*; *Prévost et al., 2015*; *Saarikangas et al., 2009*; *Sengupta et al., 2019*; *Takemura et al., 2017*; *Yandrapalli et al., 2016*), we then tested if Gag and

IRSp53 could associate directly or indirectly using immuno-precipitation (IP) assays (*Figure 2a,b*). Our results showed that IP of endogenous IRSp53 resulted in co-precipitation of Gag (*Figure 2a*, lane 1), as compared to the controls (lanes 2–4). Unfortunately, we could not assess the amount of IRSp53 pulled down by the antibody between conditions because the IgG signal masked the endogenous IRSp53 signal. To overcome this issue, we performed an IP/co-IP experiment of ectopic IRSp53-GFP/HIV-1 Gag proteins with an anti-GFP antibody, and confirmed the pull-down of IRSp53-GFP (*Figure 2b*, lanes 2 and 3) and the co-precipitation of Gag (lane 3) while nothing was detected in the controls (lanes 1 and 4). Input and flowthrough were in accordance with the results, showing less IRSp53-GFP in the flowthrough (lane 3). We then concluded that HIV-1 Gag and endogenous IRSp53, or ectopic IRSp53-GFP, were components of the same intracellular complex, interacting directly or indirectly through other factors or membrane domains. IRSp53 is a cellular protein that switches from the cytosol to the cell plasma membrane for inducing membrane ruffles upon activation by Rac1 and its effectors (*Miki and Takenawa, 2002*; *Suetsugu et al., 2006*). We have previously shown that Gag cellular expression triggers Rac1 activation (*Thomas et al., 2015*), on which IRSp53 membrane localization and function are dependent. Thus, here, we compared the relative membrane binding of IRSp53 upon cellular expression of HIV-1 Gag using membrane flotation assays (*Figure 2c,d*). In the absence of Gag ('HEK293T control' cells), one could observe the

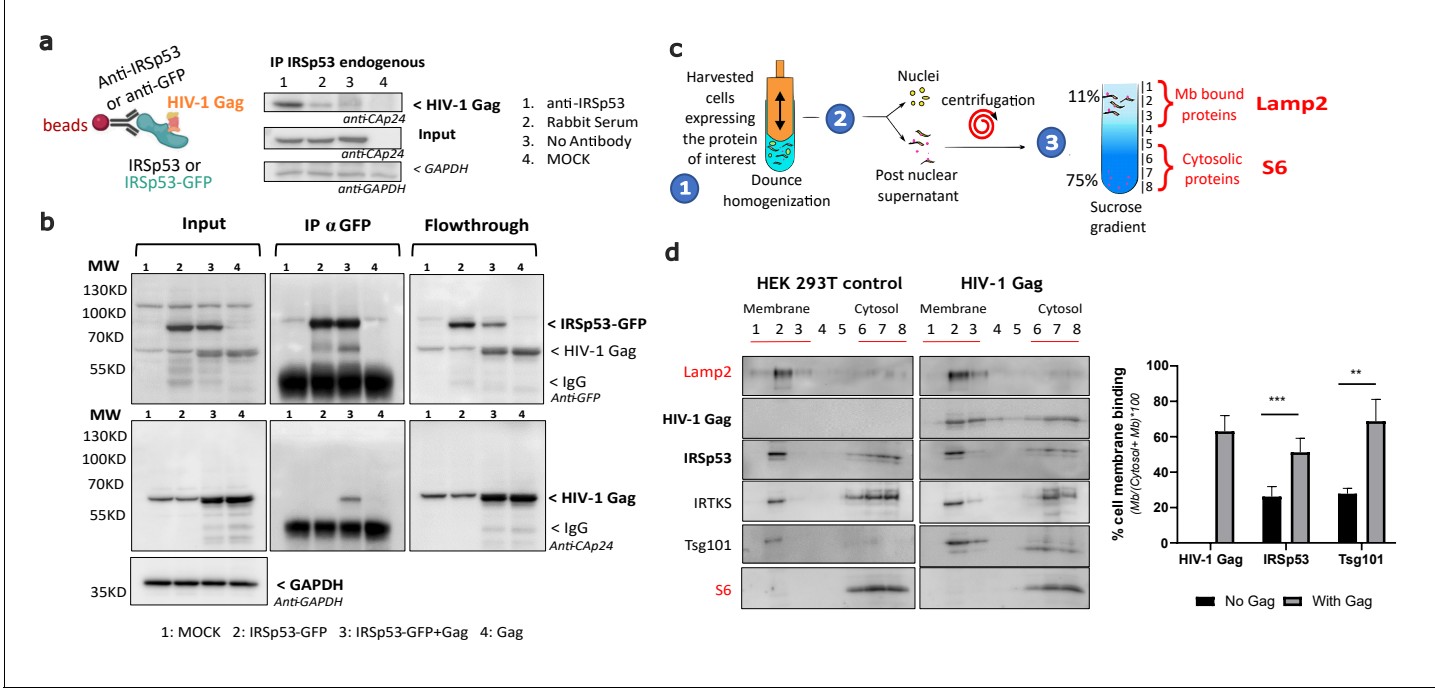

**Figure 2.** Intracellular HIV-1 Gag and IRSp53 complexation and cell membrane binding. (**a**) Co-immunoprecipitation of HIV-1 Gag/IRSp53 with an anti-IRSp53 antibody. HIV-1 Gag is enriched in the anti-IRSp53 pulldown (lane 1), as compared to the controls (lane 2: IP with an anti-rabbit serum; lane 3: no antibody; lane 4: mock, i.e., without Gag). (**b**) Co-Immunoprecipitation of HIV-1 Gag/ectopic IRSp53-GFP with an anti-GFP antibody upon overexpression of HIV-1 Gag and ectopic IRSp53-GFP in transfected HEK293 T cells. HIV-1 Gag is enriched in the anti-IRSp53-GFP pulldown (lane 3: transfected HEK293T cell lysate containing IRSp53-GFP and HIV-1 Gag), as compared to the controls (lane 1: mock without Gag or IRSp53-GFP; lane 2: IRSp53 alone; lane 4: Gag alone). Input, IP anti-GFP and flowthrough after IP are shown. (**c**) Membrane flotation assay protocol: (1) 293T cells were dounced, (2) the post-nuclear supernatant was loaded on a discontinuous sucrose gradient, and (3) following ultracentrifugation, cell membranes (lysosomal associated membrane protein, Lamp2 biomarker, fractions 1–3) were separated from the cytosolic fraction (ribosomal S6 protein biomarker, fractions 6–8). (**d**) Immunoblots of the indicated proteins (on the left) and quantification of the % of protein membrane binding in the graph below show that upon HIV-1 Gag expression in cells, IRSp53 is significantly enriched by twofold in the cell membrane fraction (p value = 0.000129; ***Student's *t*-test) (*n* = 5 independent experiments). A similar increase is observed for Tsg-101, a known interactor of the p6 domain of Gag (p value = 0.00517; **, Student's *t*-test).

The online version of this article includes the following source data and figure supplement(s) for figure 2:

**Source data 1.** Membrane flotation Quantification using Fiji for *Figure 2D* graph.

**Figure supplement 1.** Complexation of IRSp53 with Gag(i)mEOS2 is independent of Gag-p6 domain.

presence of IRSp53 both in the cytosol (*Figure 2c*, fractions 6–8, labeled with the ribosomal S6 biomarker) and in the cell membranes fractions (*Figure 2c*, fractions 1–3, labeled with the membrane Lamp2 biomarker). The lysosomal membrane protein Lamp2 and the cytosolic ribosomal protein S6 were used as controls to validate the correct separation of the membrane and cytosolic fractions. Thus, at equilibrium, 19 ± 8% of IRSp53 was bound to the cell membranes (Graph, *Figure 2d*). The same experiment was repeated with cells expressing HIV-1 Gag, where 66 ± 9% of Gag was bound to the cell membranes (Graph, *Figure 2d*). Notably, we observed a twofold increase with 44 ± 5% of IRSp53 bound to the cell membranes upon HIV-1 Gag expression (Graph *Figure 2d*) ($n$ = 5 independent experiments, $p$ value = 0.000129, Student's $t$-test), while none with IRTKS. This effect was comparable with the one of Tsg101, a protein of the ESCRT-I complex known to interact mainly with the p6 domain of Gag (*Garrus et al., 2001*; *Pornillos et al., 2003*; *von Schwedler et al., 2003*) and partially with the NC domain (*El Meshri et al., 2018*). The cellular endosomal sorting complex required for transport (ESCRT) machinery has been involved in the mechanism of vesicular budding of intracellular multi-vesicular bodies and also hijacked by the HIV-1 Gag protein for viral particle budding. Here, we observe a twofold increase in cell membrane binding of Tsg101 upon Gag expression, passing from 36 ± 10% without Gag to 79 ± 8% in the presence of Gag (*Figure 2c*) ($n$ = 5 independent experiments, $p$ value = 0.00517, Student's $t$-test).

Furthermore, we examined if Gag/IRSp53 complexation was dependent on the p6 domain of Gag to reveal if this could be independent of ESCRT recruitment by Gag. We, thus, used a C-terminal mutant of Gag, GagΔp6, which is deficient in ESCRT-Tsg101 recruitment (*von Schwedler et al., 2003*), but is still capable of binding the plasma membrane and assembling particles that poorly bud. GagΔp6 viral particles are tethered and remain attached to the plasma membrane (see *Floderer et al., 2018* for the characterization of Gag(i)mEos2Δp6). Our experiments revealed that Gag, Gag(i)mEos2, and GagΔp6(i)mEos2 were all pulled down with IRSp53 (*Figure 2—figure supplement 1*), showing that the addition of the internal mEos2 protein - a necessary tag for super resolution microscopy (SRM) imaging of Gag (see the following section) - did not affect the complexation of Gag with IRSp53. This was to show that the Gag used in the SRM imaging studies behaved similarly to wild-type Gag. Moreover, we showed that the p6 domain was not required for Gag/IRSp53 molecular interplay suggesting that it occurs before ESCRT recruitment.

Taken together, these results suggest that there is a complexation between HIV-1 Gag and IRSp53, reinforcing the idea of a strong molecular interplay between these two proteins directly or indirectly but in the same membrane domain. We observed that cellular Gag expression, possibly by triggering Rac1 activation (*Thomas et al., 2015*), favors cell membrane binding of IRSp53.

## Single-molecule localization microscopy reveals IRSp53 surrounding HIV-1 Gag assembly sites

Our finding that IRSp53 and HIV-1 Gag are present in the same molecular complex at the cell membrane motivated us to assess whether IRSp53 was present specifically at the Gag assembly sites. Because HIV-1 assembly is ~100 nm in diameter (*Floderer et al., 2018*; *Manley et al., 2008*), we used PALM (Photo-Activated Localization Microscopy) coupled to dSTORM (direct Stochastic Optical Reconstruction Microscopy) with TIRF illumination, to investigate with high precision the localization of I-BAR proteins in Gag(i)mEos2 assembly sites at the plasma membrane (*Figure 3*, *Figure 3—figure supplement 1*, *Figure 3—figure supplement 2*).

We first checked our ability to correctly identify ongoing assembly sites by quantifying the sizes of Gag clusters. For this, we used purified HIV-1 Gag(i)mEos2 VLPs (*Figure 3—figure supplement 3b*). When using the same size estimation method described in *Floderer et al., 2018*, we measured individual VLP sizes around 130–140 nm (*Figure 3—figure supplement 3b*). VLP size was correctly estimated (114 ± 37 nm, median ± 1st IQR) from the size distribution of 800 different clusters identified in our VLP images using a density-based spatial scan (DBSCAN) clustering method (*Figure 3—figure supplement 3b*). Finally, DBSCAN showed us that these HIV-1 Gag(i)mEos2 cluster size distributions had similar median values for each cell type in each condition (HEK 293T (96 ± 44 nm, *Figure 3—figure supplement 3c*) or Jurkat T cells (105 ± 66 nm, *Figure 3—figure supplement 3e*) when IRSp53 was immunolabelled, HEK 293T (116 ± 68 nm, *Figure 3—figure supplement 3d*) or Jurkat T cells (96 ± 50 nm, *Figure 3—figure supplement 3f*) when IRTKS was immunolabelled), allowing us to directly compare IRSp53 and IRTKS organization close to these assembly sites.

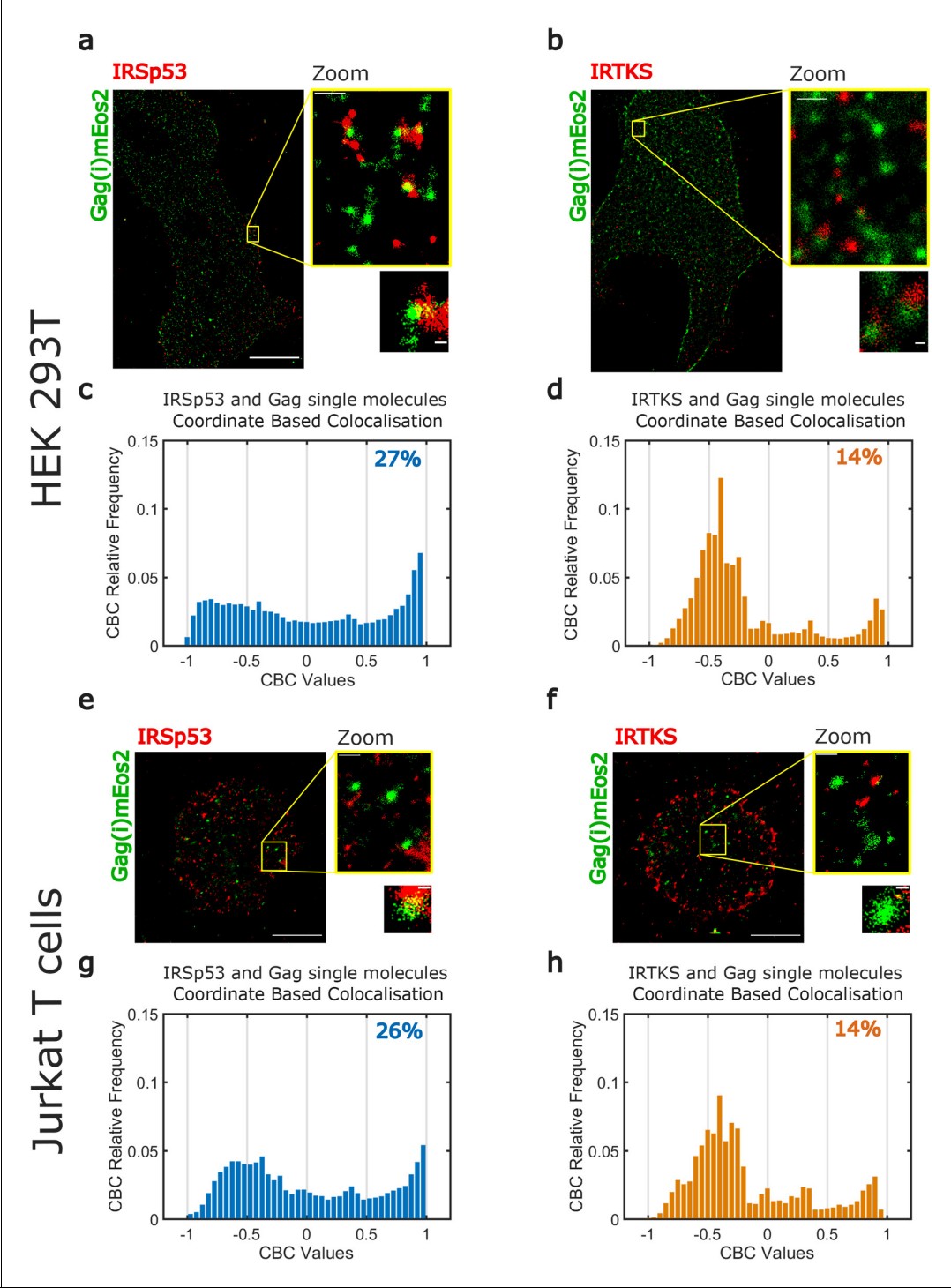

**Figure 3.** Super-resolution microscopy imaging reveals preferential IRSp53 localization at HIV-1 Gag budding sites. (a, b) Super-resolved PALM/STORM dual-color images of HEK293T cells expressing Gag(i)mEos2 (green) and immunolabeled for IRSp53 (red, a) or IRTKS (red, b) (scale bar = 10 µm) with a magnified view (scale bar = 500 nm) and selected zoom-in images of single Gag(i)mEos2 clusters (scale bar = 100 nm). (c, d) Quantification of coordinate-based colocalization (as in *Malkusch et al., 2012*) at Gag assembly sites: CBC values for IRSp53 (c) and IRTKS (d) were plotted as relative frequencies. IRSp53 CBC distribution (c) shows that 27% of all IRSp53 localizations are highly correlated with Gag(i)mEos2 localizations (>0.5). On the other hand, only 14% of IRTKS localizations (d) are highly correlated, while IRTKS CBC distribution exhibits a peak of anti-correlated/non-correlated (−0.5 to 0) localization. (e, f) Super-resolved dual-color PALM/STORM images of Jurkat T cells expressing Gag(i)mEos2 (green) and immunolabeled for IRSp53 (red, e) or IRTKS (red, f) with a magnified view (scale bar = 500 nm) and single Gag(i)mEos2 cluster zoom (scale bar = 100 nm). (g, h) Relative frequency distribution plot of the CBC values for IRSp53 (g) and IRTKS (h). (g) shows that 26% of IRSp53 localizations are highly correlated with Gag(i)

*Figure 3 continued on next page*

*Figure 3 continued*

mEos2 localizations (>0.5) while (**h**) shows that IRTKS localisations are mainly anti-correlated to non-correlated (−0.5 to 0) with Gag(i)mEos2 localizations.

The online version of this article includes the following source data and figure supplement(s) for figure 3:

**Source data 1.** Experimental data CBC for IRSp53.
**Source data 2.** Experimental data CBC for IRTKS.
**Source data 3.** cluster sizes for IRSp53.
**Source data 4.** cluster sizes for IRTKS.
**Figure supplement 1.** Super-resolution PALM/STORM images of HEK293T cells expressing HIV-1 Gag(i)mEos2 (in green) and immunolabeled for (**a**) IRSp53 (Atto647N, in red) and (**b**) IRTKS (Atto647N, in red), with magnified images adjacent to the main image.
**Figure supplement 2.** Super-resolution PALM/STORM images of Jurkat T cells expressing HIV-1 Gag(i)mEos2 (in green) and immunolabelled for (**a**) IRSp53 (Atto647N, in red) and (**b**) IRTKS (Atto647N, in red), with magnified images adjacent to the main image.
**Figure supplement 3.** Localization precision and size determination of HIV-Gag clusters.
**Figure supplement 4.** Workflow of image analysis for PALM/STORM images.

In HEK293T and Jurkat T cells, reconstructed dual-color PALM/STORM images exhibited Gag(i)mEos2 assembly sites close to or overlapping with IRSp53 (*Figures 3a,e*, *Figure 3—figure supplement 1a*, *Figure 3—figure supplement 2a*) whereas Gag clusters did not seem to overlap with IRTKS (*Figures 3b,f*, *Figure 3—figure supplement 1b*, *Figure 3—figure supplement 2b*). These results are consistent with the siRNA data presented in *Figure 1b*. In order to quantify these observations, we performed coordinate-based colocalization (*Malkusch et al., 2012*) (CBC) analysis of HIV-1 Gag and IRSp53 (or IRTKS) in close proximity of HIV-1 assembly sites. We first isolated the assembly sites by segmentation and retrieved their center positions. We then kept all HIV-1 Gag located within a distance of 80 nm from this center (70–80% of all cluster sizes measured by DBSCAN are found within this distance) and all the IRSp53 (or IRTKS) found in a distance of 150 nm from this center (~twofold the assembly site size, see *Figure 3—figure supplement 4* for details on the process workflow). We chose this IRSp53 (or IRTKS) cutoff distance to minimize the contribution of cross-colocalization between different HIV-1 Gag clusters. In contrast to classical colocalization analysis, CBC takes into account the spatial distribution of biomolecules to avoid excessive colocalization due to local densities of single molecules and provides a colocalization value for each single-molecule localization. This CBC value ranges from −1 to +1, where −1 corresponds to anti-correlation, 0 indicating non-correlation, and +1 corresponds to perfect correlation between the two molecules. Since CBC values are calculated for each localization, we plotted the CBC values as frequency distributions for all localizations of IRSp53/Gag and IRTKS/Gag. As shown in *Figure 3c* (for *n* = 4 HEK293T cells) or in *Figure 3g* (for *n* = 5 Jurkat T cells), the CBC distribution for IRSp53/Gag has a higher proportion of values exhibiting high colocalization (27% in HEK293T cells or 26% in Jurkat T cells of CBC > 0.5) in comparison to IRTKS/Gag values (14% for both cell types, *n* = 5 for HEK293T and Jurkat T cells of CBC > 0.5) which instead show a very high proportion (close to 75%) of anti-correlation (CBC < 0) (*Figure 3d,h*). This comparison directly shows that IRSp53 displays stronger single-molecule colocalization with Gag in assembling clusters than IRTKS does.

Although CBC values give a quantitative value of the colocalization, it does not provide direct information on the average positions of IRTKS or IRSp53 molecules with respect to Gag molecules within the assembling clusters. Moreover, as for all the colocalization methods used to analyze molecule proximities, the CBC method is a strongly parameterize method, the results of which can depend on the parameter values (total distance of search for colocalization, number of searching circles...). Thus, to gain more insight into these colocalization quantification, we performed simulations to generate different patterns of PALM/STORM localizations (*Figure 4*), and analyzed them with the same set of parameters (total distance and distance steps) that the one we used for the experimental data. Since IRSp53 and IRTKS are membrane curvature sensors and since we observed that a lack of IRSp53 stops the assembling bud at half completion (*Figure 1e*), we hypothesis that they will be located around the ongoing assembly site. We therefore simulated an annulus of IRSp53 or IRTKS STORM localizations with different waists located at increasing distances from the center of Gag clusters (*Figure 4a* and *Figure 4—figure supplement 1*). Numerically generated images were then analyzed identically to our experimental images (see 'Materials and methods' for details) and CBC distributions were generated (*Figure 4b*). Experimental and simulated cumulative distribution

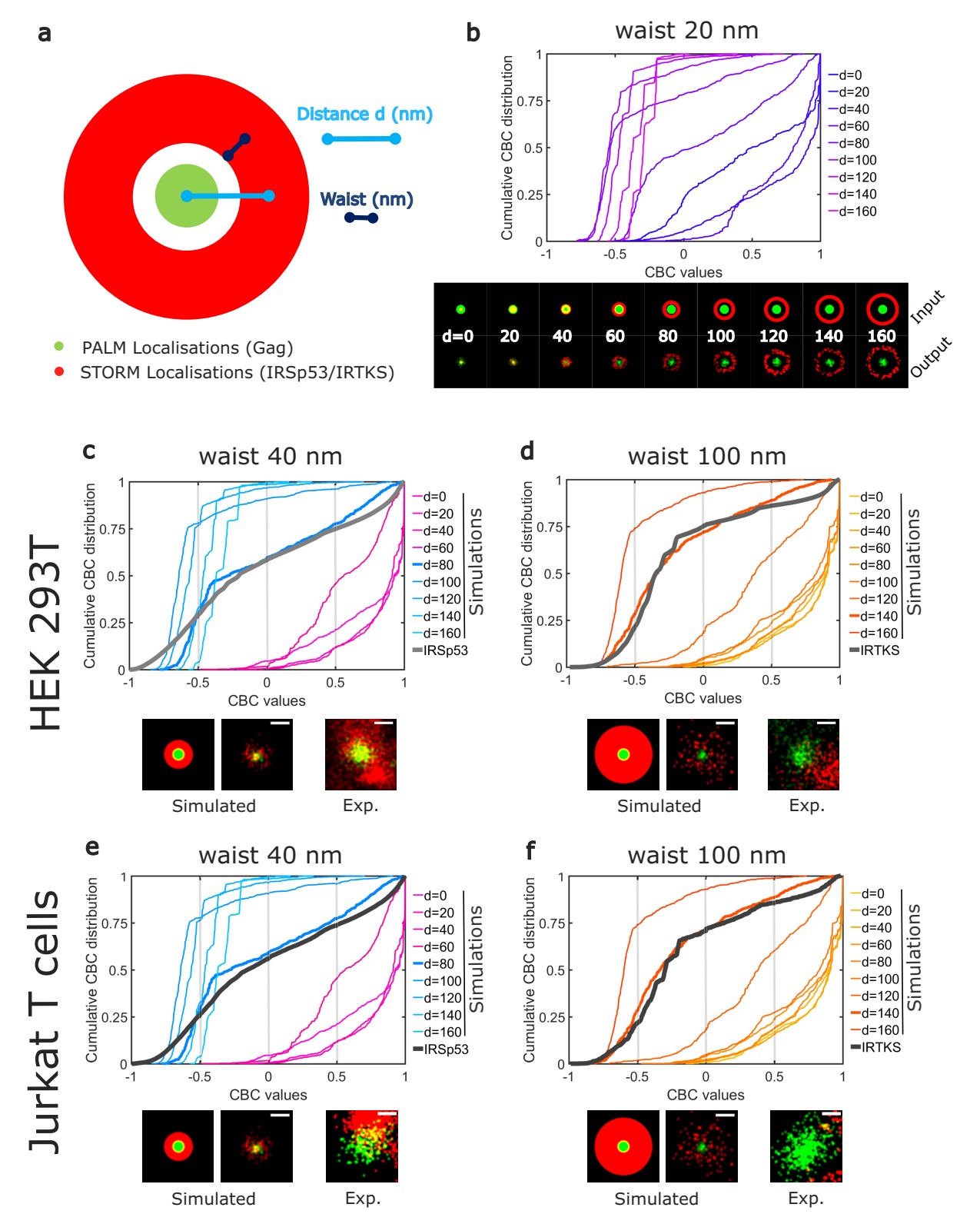

**Figure 4.** Numerical simulations confirm preferential IRSp53 localization around HIV-1 Gag budding sites. (a) Schematic representation of the different configurations used in the numerical simulation mimicking belts of given waists (w) of IRSp53 or IRTKS localizations (red) surrounding an HIV-1Gag(i) mEos2 assembly site (green) at a given distance ( d). (b) Numerically simulated cumulative distribution function of CBC values obtained for a 20 nm waist belt of IRSp53 or IRTKS at different distances from 0 to 160 nm (top). Below are found the schematic representations (upper part, input) and the

*Figure 4 continued on next page*

*Figure 4 continued*

ThunderSTORM reconstructed images obtained from the simulated positions (lower part, output, see 'Materials and methods' for details) for the 20 nm waist belt at different positions. (c, d, e, f) Experimental CBC values for IRSp53 in HEK (c) or Jurkat T cells (e) and IRTKS in HEK (d) and Jurkat T cells (f) were plotted as cumulative frequency distributions and compared to simulated distributions obtained at different distances and structures (see *Figure 4—figure supplement 1* for the whole data sets). For both cell type, IRSp53 shows a cumulative CBC distribution corresponding to a simulation with a waist of 40 nm (width 80 nm) at a distance of 80 nm (left graph, bold gray lines correspond to the experimental data for IRSp53, bold blue lines correspond to the simulated values closest to experimental data, see *Figure 4—figure supplement 1g,h,i,j* for statistics). IRSp53 thus corresponds to a restricted pattern in and around a Gag assembly site (panel 1 schematic of simulated data, panel 2 simulated data, and panel 3 experimental data). On the other hand, the IRTKS experimental CBC distribution (bold gray line in the graph) is similar to simulations with a waist of 100 nm (width 200 nm) at a distance of 140 nm (bold red line). IRTKS is more diffuse and spreads out (panel 1 schematic of simulated data, panel 2 simulated data, and panel 3 experimental data). Scale bar in the panels = 100 nm.

The online version of this article includes the following source data and figure supplement(s) for figure 4:

**Source data 1.** simulation-CBC-dataset-1 for IRSp53.
**Source data 2.** simulation-CBC-dataset-2 for IRTKS.
**Figure supplement 1.** Cumulative frequency distributions of CBCs for simulated PALM/STORM data.

functions of the CBC values were compared by performing a root mean square error (RMSE) quantification (*Figure 4—figure supplement 1, g,h,i,j*). The lowest RMSE value was considered as the best similarity between the two distributions. This comparison indicated that IRSp53 localization, on average, displays a restricted pattern around and in the assembly sites. This corresponds to a circular ring surrounding the assembly site at 80 nm from the center of the Gag budding sites with a width of 80 nm (*Figure 4c,e*). On the other hand, IRTKS was present as a large diffuse pattern located at 140 nm from the Gag assembly site center with a width of 200 nm, explaining why fewer IRTKS molecules were detected in the assembly sites (*Figure 4d,f*). Interestingly, the maps of the RMSE values show that, on average, the surrounding belt center (independently of its waist) is located nearer to the HIV-1 Gag assembly center in the case of IRSp53 as compared to that of IRTKS (*Figure 4—figure supplement 1g,h,i,j*). Our results thus show that IRSp53 indeed specifically localizes at HIV-1 Gag assembly sites at the cell plasma membrane, whereas IRTKS poorly does.

The involvement of IRSp53 around Gag assembly sites seems to be conserved regardless of the cell type, reinforcing the idea of a specific role for IRSp53 in HIV-1 Gag particle assembly.

## IRSp53 is incorporated in HIV-1 particles

To assess IRSp53 incorporation into HIV-1 Gag particles, we purified Gag-VLPs from cells transfected with Gag-mCherry and several GFP-tagged I-BAR domain proteins (*Figure 5a*). The IRSp53-I-BAR-GFP construct only contains the membrane curving I-BAR domain of IRSp53. PH-PLCδ-GFP, a PI(4,5) $P_2$ binding protein, was used as a control, because it binds PI(4,5)$P_2$ but does not generate membrane curvature. Fluorescent VLPs were purified from these transfected cells, then visualized for two colors (green: GFP and red: mCherry), and then a Mander's coefficient was calculated as an indicator of incorporation of the ectopic (green) GFP-tagged proteins within the (red) Gag-mCherry VLPs and vice-versa (see Materials and Methods) (*Figure 5a,b*). We found a high correlation (Mander's coefficient = 0.95–1) between IRSp53-GFP and Gag-mCherry (*Figure 5b*, left graph, red column), indicating that almost all Gag-mCherry VLPs contained IRSp53-GFP, while the reverse (green column) was not true, indicating that all the IRSp53-GFP-labelled vesicles produced by the cells were not all positive for Gag. When using the IRSp53-I-BAR domain alone, we also obtained a high Mander's coefficient, that is, ~0.8 (*Figure 5b*). In contrast, for IRTKS-GFP, the Mander's coefficient was 0.4–0.5, indicating no significant correlation between IRTKS-GFP and Gag-mCherry. This indicates that even if a domain (PH-PLCδ) or another I-BAR protein (IRTKS) can recognize PI(4,5)$P_2$ at the cell membrane, it is not sufficient for incorporation into Gag-VLPs. Finally, we found that the reverse correlation of GFP-tagged proteins with Gag-mCherry was under the random 0.5 coefficient (*Figure 5b*, right graph, green column), indicating that, on average, GFP-tagged vesicles produced by the cells did not contain Gag. Taken together, these results show the preferential incorporation of IRSp53-GFP into released HIV-1 Gag-mCherry VLPs.

To study the incorporation of endogenous IRSp53 in HIV-1 particles, cells were transfected with plasmids expressing either wild-type infectious HIV-1 or codon-optimized immature HIV-1 Gag protein (without genomic RNA). The virus particles were purified through a 20%-sucrose cushion or

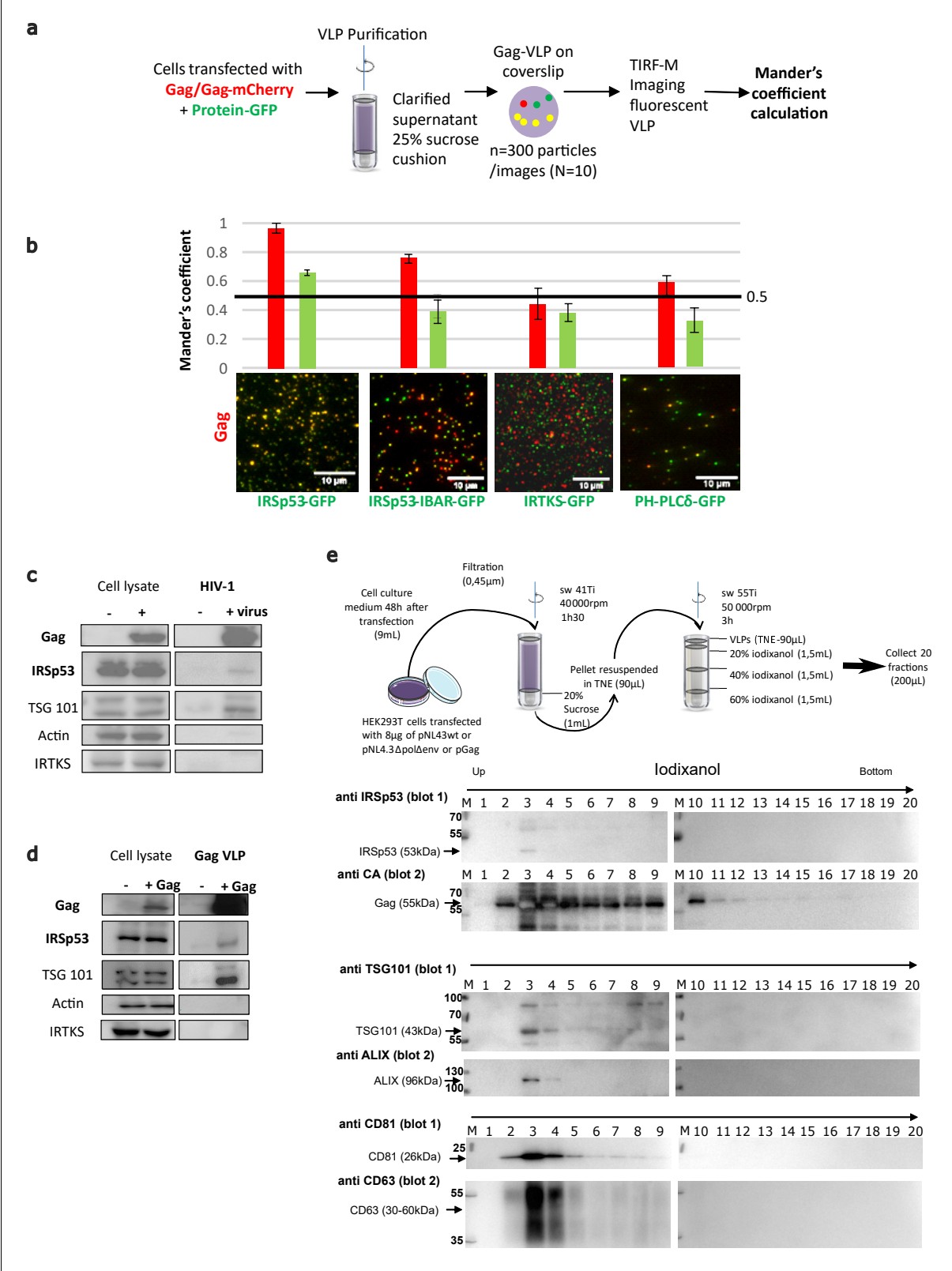

**Figure 5.** IRSp53 is incorporated into HIV-1 particles in a Gag-dependent manner. (**a**) Schematic for the protocol followed for imaging and analysis. HIV-1 Gag VLPs were purified from HEK293T cells expressing HIV-1 Gag/Gag-mCherry and IRSp53-GFP, or other GFP tagged proteins binding PI(4,5)P$_2$ (IRSp53-IBAR-GFP, IRTKS-GFP, and PH-PLCδ-GFP). Purified Gag VLPs were then spotted over a poly-lysine treated glass slide and imaged by TIRF-Microscopy (particles were imaged in the red, and IRSp53 in the green channel). For each condition, 3000 particles (~300 particles/image, 10 images)

*Figure 5 continued on next page*

*Figure 5 continued*

were counted. Fluorescence correlation (Mander's coefficient, see Materials and Methods for details) was determined for Gag-mCherry and for IRSp53-GFP, IRTKS-GFP, and PH-PLCδ-GFP and reported in the graphs. (b) The 0.5 value indicates a random incorporation level (indicated by black line across the graph). IRSp53-GFP and IRSp53-IBAR show high correlation values (0.95–1 and 0.8, respectively). The other I-BAR domain proteins were not significantly correlated with Gag-mCherry particles (0.4–0.5). PH-PLCδ-GFP, a known marker of the phospholipid PI(4,5)P$_2$, shows a slightly higher correlation (0.6), since HIV-1 Gag is known to associated with this phospholipid. (c) Incorporation of IRSp53 into wild-type pNL4-3 HIV-1 or (d) Gag VLPs revealed by immunoblots against Gag(p24), IRSp53, IRTKS, Tsg101, or actin, as indicated. Following a 25% sucrose cushion purification, IRSp53 was found to be associated with released wild-type HIV-1 (left panel) and Gag VLPs (right panel). Tsg101, known to be incorporated into released particles, was also associated with viral particles. IRTKS, a closely related I-BAR protein to IRSp53, was not incorporated in purified HIV-1 viral particles or Gag-VLPs. (e) Protocol of VLPs purification using sucrose cushions and an iodixanol gradient. Briefly, pellets obtained after ultracentrifugation of cell culture medium of HEK293T transfected with pNL4.3HIV-1 or pGag were deposed on an iodixanol gradient (20%, 30%, and 60%). 20 fractions of 200 μL were collected from the top of the tube. Fractions collected following an iodixanol gradient purification of NL4-3ΔPolΔEnv Gag VLPs were analyzed using Western blots for IRSp53 and Gag, TSG101 and ALIX, CD81 and CD63 revealed respectively on the same membrane (blots 1 and 2) revealing IRSp53 association with Gag viral particles and known cofactors.

further through a continuous iodixanol density gradient (as in *Grigorov et al., 2009* and *Grigorov et al., 2006*). IRSp53 was found to be associated with the viral particles in both conditions, that is, in infectious HIV-1 and in Gag VLPs, indicating that Gag alone is sufficient to recruit IRSp53 in the viral particles (*Figure 5c,d*, 'IRSp53'). Tsg101 also showed an association with viral particles in both conditions (*Figure 5c,d*, 'Tsg101'), as reported previously (*Garrus et al., 2001*; *Hammarstedt and Garoff, 2004*; *Pornillos et al., 2003*). In contrast, IRTKS was not associated neither with Gag-VLP nor HIV-1 particles (*Figure 5c,d*, 'IRTKS'). Upon further purification (*Figure 5e*), IRSp53, and the ESCRT proteins, Tsg101 and ALIX, were found to be associated within the same fractions containing the HIV-1 Gag viral particles, together with other well-known viral particle cofactors such as CD81, CD63 tetraspanins (*Grigorov et al., 2009*; *Grigorov et al., 2006*). Thus, endogenous IRSp53 is most probably incorporated in HIV-1 particles in a Gag-dependent manner.

## HIV-1 Gag is enriched at membrane tube tips generated by IRSp53 I-BAR domain

The results above demonstrate that not only is IRSp53 incorporated in Gag-VLPs, but it is present at the budding sites and its deletion strongly reduces HIV-1 particle release in a Gag-dependent manner by arresting its bud assembly at the cell plasma membrane (*Figure 1*). In order to advance molecular mechanistic understanding of the role of IRSp53 locally at Gag assembly sites, we assessed IRSp53 I-BAR/Gag interplay on model membranes giant unilamellar vesicles (GUVs). Previous in vitro studies showed that when placing IRSp53 I-BAR domain outside PI(4,5)P$_2$-containing GUVs, the I-BAR domain can deform GUV membranes, generating tubes toward the vesicle interior (*Jarin et al., 2019*; *Prévost et al., 2015*; *Saarikangas et al., 2009*; *Jarin et al., 2021*). Consistent with the previous observations, we observed I-BAR driven tubulations on GUVs at low I-BAR concentrations (0.005–0.06 μM, see *Figure 6*, *Figure 6—figure supplement 1c* as an example); surprisingly, when increasing I-BAR concentrations to 0.1 μM and up to 1 μM, we observed a decrease in the number of GUVs having tubes. Future work is required to investigate this seemingly puzzling observation.

For our Gag membrane binding assay, we first used a high concentration of IRSp53 I-BAR domain (0.5 μM), while keeping the PI(4,5)P$_2$ concentration constant. This was done in order to prevent the generation of tubes by IRSp53 I-BAR domain and to focus on analyzing the membrane-binding efficiency of Gag in the presence of IRSp53 I-BAR domain on *flat*, non-deformed GUVs. We found that Gag binding to GUV membranes is increased ~7-fold when IRSp53-I-BAR domain was introduced first on GUVs before adding Gag (median value 6.7), compared to the condition of Gag only (median value 0.9) ($p$ <0.0001, Student's $t$-test) (*Figure 6a*, 'Gag only' vs. 'I-BAR + Gag'). However, in the condition where Gag was introduced before adding the I-BAR domain, Gag intensity on GUV membranes increased only ~3-fold as compared to the Gag only condition ($p$ < 0.0001, Student's $t$-test) (*Figure 6a*, 'Gag only' vs. 'Gag + I-BAR'). Notably, by comparing I-BAR + Gag and Gag + I-BAR conditions, we observed a twofold higher Gag intensity on GUV membranes in the first condition ($p$ < 0.0001, Student's $t$-test) (*Figure 6a*, 'I-BAR + Gag' vs. 'Gag + I-BAR'). Taken together, these results show that IRSp53 I-BAR domain facilitates Gag membrane binding on GUV in favor of

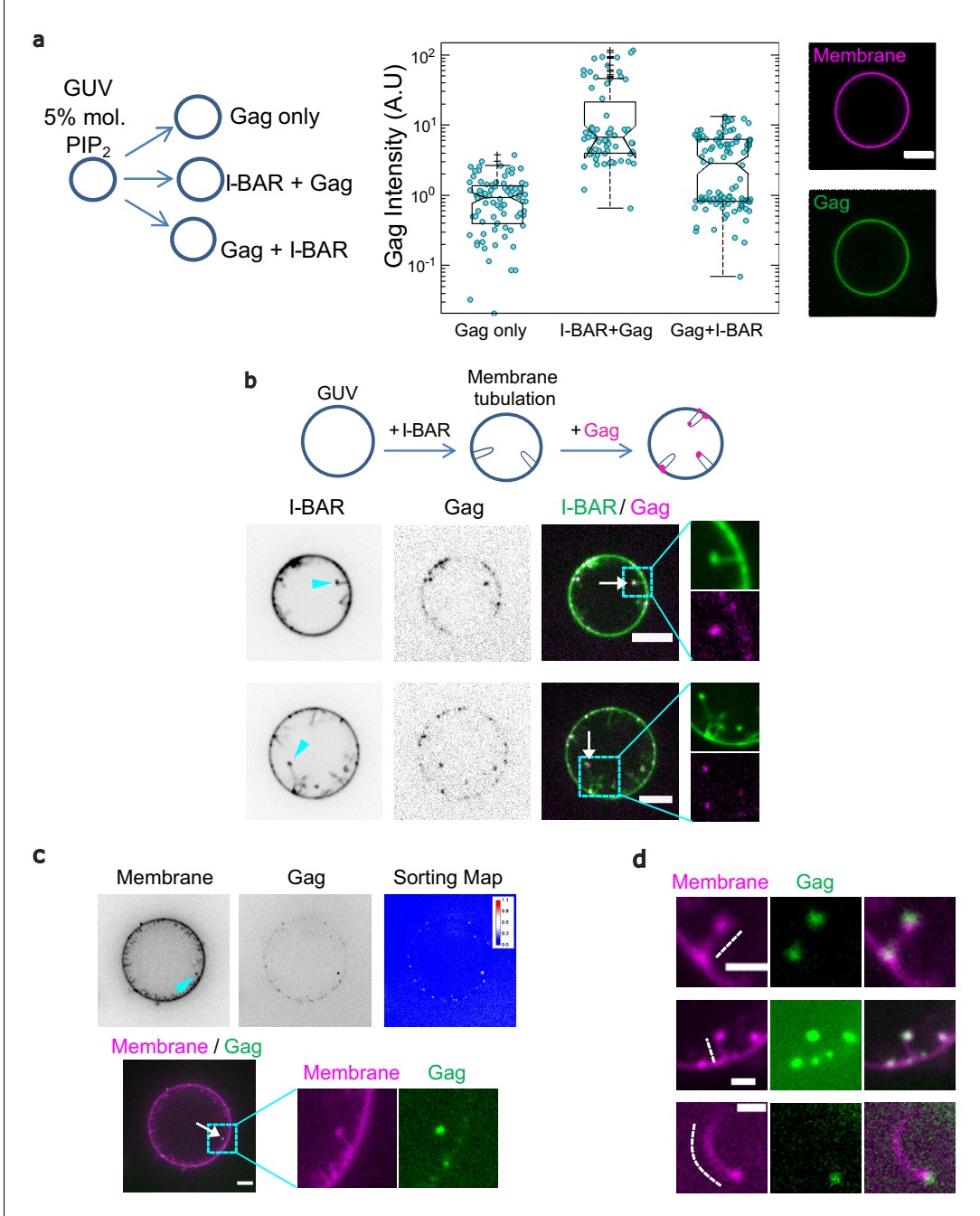

**Figure 6.** IRSp53 I-BAR domain enhances Gag recruitment to GUV-membranes and at the tip of I-BAR domain-induced tubes. (a) (Left) AX488 Gag fluorescence intensity on membranes in the absence of I-BAR domain (named 'Gag only'), in the presence of I-BAR domain where GUVs were first incubated with I-BAR domain and then Gag (named 'I-BAR + Gag') and GUVs were first incubated with Gag and then I-BAR domain (named 'Gag + I-BAR'). Each circle presents one GUV analysis. $N = 82$ GUVs, $n = 4$ sample preparations for 'Gag only,', $N = 67$ GUVs, $n = 4$ sample preparations for 'I-BAR + Gag,' and $N = 104$ GUVs, $n = 4$ sample preparations for 'Gag + I-BAR'. To pool all data points from the four sample preparations, in each preparation for all three conditions, Gag intensities were normalized by the mean Gag intensity in the 'Gag only' condition. Protein bulk concentrations: 0.3 μM for AX488 Gag and 0.5 μM for I-BAR domain (not fluorescently labeled). (Right) Representative confocal images of AX488 Gag on GUV membranes in 'I-BAR + Gag' condition. To visualize GUV membranes, 0.5 mol% of BODIPY-TR-C5-ceramide was incorporated in the membranes. (b) Representative confocal images of AX594 Gag in I-BAR domain-induced tubules. Inverted grayscale images are shown for I-BAR domain and Gag. Protein bulk concentrations: 0.3 μM for AX594 Gag and 0.05 μM for I-BAR domain (70% unlabeled and 30% AX488 labeled I-BAR domain). Cyan arrowheads point out I-BAR domain-induced tubules and white arrows indicate the colocalization of Gag and I-BAR domain at the tips of the tubules. (c and d) Representative confocal images of AX488 Gag (green) in I-BAR domain-induced tubules. Protein bulk concentrations: in (c) 0.1 μM for AX488 Gag and 0.05 μM for I-BAR domain (not fluorescently labeled); in (d) 0.3 μM for AX488 Gag and 0.05 μM for I-BAR domain (not fluorescently labeled).

*Figure 6 continued on next page*

*Figure 6 continued*

To visualize GUV membranes, 0.5 mol% of BODIPY-TR-C5-ceramide (magenta) was incorporated in the membranes. In (c), inverted grayscale images were shown for membranes and Gag. The cyan arrowhead points out an I-BAR domain-induced tubule and white arrow indicates Gag signals at the tip of the tubule. Sorting map was obtained by calculating the fluorescence intensity ratio of Gag and membranes (see Material and Methods for more details). In (d), dashed white lines indicate I-BAR domain-induced tubules. Scale bars, (a–c) 5 µm and (d) 2 µm. GUV, giant unilamellar vesicle.

The online version of this article includes the following figure supplement(s) for figure 6:

**Figure supplement 1.** In vitro interplay between IRSp53 I-BAR and HIV-1 Gag binding to GUV.

**Figure supplement 2.** Dotty signals of Gag in GUVs with I-BAR induced tubules.

increasing Gag concentration locally. Furthermore, given that in our cell experiments, we observed a relative increase of IRSp53 bound to the cell membranes upon HIV-1 Gag expression (*Figure 2d*), we also assessed if HIV-1 Gag could facilitate the membrane binding of IRSp53 I-BAR domain. Consistent with our cell experiment results, we observed a similar increase of the membrane binding (~1.5-fold) of IRSp53 I-BAR domain in the presence of HIV-1 Gag on GUVs (*Figure 6—figure supplement 1a,b*) suggesting a strong interplay between Gag and IRSp53 I-BAR domain.

Given these results, and that IRSp53 is a membrane curving protein involved in the early stages of cell protrusion generation (*Disanza et al., 2013*; *Sathe et al., 2018*), we asked whether the local membrane deformation induced by IRSp53 could be a preferred location for HIV-1 Gag assembly. We incubated GUVs with IRSp53 I-BAR domain at a low concentration (0.05 µM), which allows for the generation of inward membrane tubes, followed by the addition of Gag (*Figure 6b*, see Materials and Methods). This experiment revealed that Gag was sorted preferentially to the tips of the tubes generated by the IRSp53 I-BAR domain (*Figure 6b,c,d*, *Video 1* and *Figure 6—figure supplement 1c,d*). We note that the majority of the tubes in GUVs were moving too rapidly, preventing us from acquiring images with high spatial resolution (see *Videos 2* and *3*). However, we observed that the Gag signals appeared dotty inside GUVs (*Figure 6—figure supplement 2* and *Videos 2* and *3*), which is very different from the IRSp53 I-BAR domain signal that is clearly all along the tubes (*Figure 6—figure supplement 1c*). Moreover, for tubes that were not moving too fast, we found that for all the tubes (17 tubes, protein concentrations: 0.05 µM unlabeled I-BAR domain and 0.3 µM AX488 Gag), the Gag signal was exclusively located at the tips of the tubes (*Figure 6d* and *Video 1*). Finally, we observed that addition of HIV-1 Gag resulted in the formation of shorter I-BAR tubules as compared to GUVs incubated with the IRSp53-I-BAR domain alone (*Figure 6—figure supplement 1c,d*), indicating an interference in I-BAR tubule elongation when Gag sorted to the tubule tips, suggesting that Gag usurps the IRSp53 tubulation function.

Taken together, these results demonstrate that HIV-1 Gag binding to membranes is enhanced locally by the presence of IRSp53 I-BAR domain, and that Gag preferentially binds to highly curved membranes generated by the I-BAR domain of IRSp53.

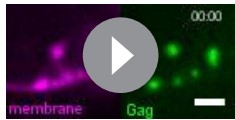

**Video 1.** Imaging of an I-BAR domain-induced tubule having Gag signal at its tip. Time-lapse imaging of an I-BAR domain-induced tubule. GUVs were first incubated with IRSp53 I-BAR domain (0.05 µM, unlabeled), followed by addition of HIV-1 Gag (0.3 µM, AX488 labeled, green). GUV membranes contained 0.5 mol% of BODIPY-TR-C5-ceramide (magenta). Frame interval = 0.6 s. Time in mm:ss. Scale bar = 2 µm. GUV, giant unilamellar vesicle.
https://elifesciences.org/articles/67321#video1

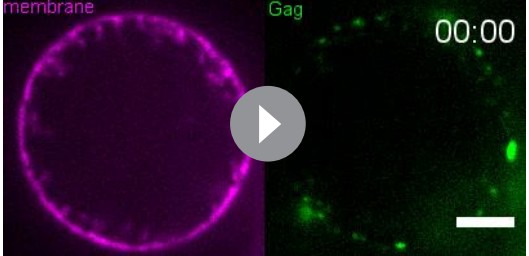

**Video 2.** Imaging of a GUV having I-BAR domain-induced tubules. GUVs were first incubated with IRSp53 I-BAR domain (0.05 µM, unlabeled), followed by addition of HIV-1 Gag (0.3 µM, AX488 labeled, green). GUV membranes contained 0.5 mol% of BODIPY-TR-C5-ceramide (magenta). Frame interval = 0.6 s. Time in mm:ss. Scale bar = 5 µm. GUV, giant unilamellar vesicle.
https://elifesciences.org/articles/67321#video2

## Discussion

The findings of this study uncovered the role of the host cellular I-BAR factor IRSp53 in HIV-1 Gag assembly and membrane curvature upon bud formation. In vitro, on GUVs, we showed that the IRSp53 I-BAR domain enhances Gag membrane binding locally (*Figure 6*), and vice versa (*Figure 6—figure supplement 1a,b*), in agreement with cell membrane flotation assays also showing an increase of IRSp53 membrane retention upon Gag expression (*Figure 2*). Indeed, IRSp53 was found at, or in close vicinity to Gag assembly platforms at the cell membrane (*Figures 3* and *4*), and is incorporated into Gag-VLPs and in HIV-1 virions (*Figure 5*). Importantly, we revealed that the partial knockdown of IRSp53 gene expression arrested Gag assembly at the mid-bud formation stage (*Figure 1*) and that Gag preferentially localizes at the tube tips induced by IRSp53 I-BAR domain, interfering with its long tubule formation in vitro (*Figure 6*). Altogether, IRSp53 appears instrumental in membrane curvature upon HIV-1 budding and is locally subverted as an essential factor needed for full HIV-1 Gag particle assembly.

Using GUVs, we observed that Gag not only colocalizes with IRSp53 I-BAR domain on vesicles, but that the IRSp53-I-BAR domain increases Gag binding to these model membranes, mimicking the possible local Gag/IRSp53 interplay at the assembly site (*Figure 6a*). Indeed, BAR domain proteins, in general, and IRSp53 in particular, are known to induce strong $PI(4,5)P_2$ clusters (*Saarikangas et al., 2009*; *Zhao et al., 2013*), $PI(4,5)P_2$ was shown to play a role in Gag binding to the cell plasma membrane (*Ono et al., 2004*), as well as $PI(4,5)P_2$ is strongly clustered during virus assembly (*Favard et al., 2019*; *Ono et al., 2004*; *Yandrapalli et al., 2016*). Thus, these results suggest that the membrane binding of Gag on IRSp53-enriched membrane domains could promote the plasma membrane binding of both proteins (*Figure 6*). This is in agreement with super-resolution imaging in cells, where Gag/IRSp53 interactions may take place at the Gag assembly sites as IRSp53 was localized in close proximity to Gag assembly sites in both HEK293T cells and CD4 Jurkat T cells (*Figure 3*). Our experiments suggest that Gag and IRSp53 are associated in a common complex at the cell plasma membrane (*Figure 2*). Given that upon Gag expression, IRSp53 increases its binding to the cell membrane (*Figure 2*), this suggests that Gag could activate IRSp53 through Rac1 activation (*Thomas et al., 2015*) or perhaps by releasing its auto-inhibition (*Kast et al., 2014*). However, these explanations remain to be tested.

HIV-1 particles are known to incorporate a large number of cellular proteins, many of which are directly involved in virus budding (*Hammarstedt and Garoff, 2004*). Here, we showed that IRSp53 is incorporated in Gag-VLPs, as well as in purified HIV-1 virions (*Figure 5*), which most likely depends on the I-BAR domain of IRSp53 (*Figure 5*). Using the IRSp53-I-BAR domain on GUVs, we induced membrane protrusions that have a negative mean curvature similar to a viral bud; Gag was found particularly at the tube tips that have a half-sphere geometry similar to a viral bud (*Figure 6*, *Videos 1*, *2* and *3*). This indicates that Gag binds preferentially to IRSp53 I-BAR-curved membranes in vitro in contrast to other almost-flat areas of the GUVs. Similarly, in cells, single-molecule localisation images reveal some Gag clusters enriched at IRSp53 labeled protrusions at the plasma membrane (*Figure 3—figure supplement 1*). IRSp53 clusters have already been reported prior to filopodia formation (*Disanza et al., 2013*) and in negatively curved area at the onset of endocytic buds (*Sathe et al., 2018*). Moreover, it was shown that inducing local membrane curvature helps to initiate Gag lattice formation (*Pak et al., 2017*). We thus propose that IRSp53 induces local membrane curvature, most likely upon activation through Rac1/Cdc42 and effectors, which in turn can promote local Gag recruitment and initiation of the viral assembly (knowing that expression of Gag can activate Rac1, *Thomas et al., 2015*). This appears to be independent of the cell types (*Figure 3*).

Although the presence of RNA can facilitate the growth of the Gag network (*Chen et al., 2014a*; *Floderer et al., 2018*), favoring membrane bending due to the intrinsic curvature of assembling Gag hexamers, coarse-grained simulations of HIV-1 Gag assembly showed that, above a certain threshold, this Gag self-assembly is unable to overcome the free energy penalty required to curve the membrane. Here, we observed that siRNA knockdown of IRSp53 gene expression induces a decrease in viral particle production and arrests the assembly at half completion (*Figure 1*). Since IRSp53 stabilizes curvature by scaffolding (*Prévost et al., 2015*), another role of IRSp53 could be to lower this free energy barrier involved in the progression of the budding process beyond the half-sphere geometry by stabilizing long enough the bud curvature. This stabilization could be achieved either directly by organizing linearly around the assembly site (*Jarin et al., 2019*) and mechanically

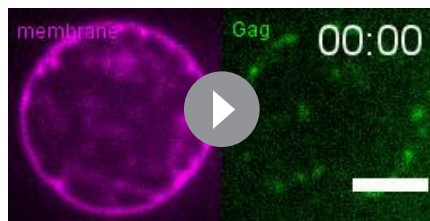

**Video 3.** Imaging of a GUV having I-BAR domain-induced tubules. GUVs were first incubated with IRSp53 I-BAR domain (0.05 µM, unlabeled), followed by addition of HIV-1 Gag (0.3 µM, AX488 labeled, green). GUV membranes contained 0.5 mol% of BODIPY-TR-C5-ceramide (magenta). Frame interval = 0.6 s. Time in mm:ss. Scale bar = 5 µm. GUV, giant unilamellar vesicle.

https://elifesciences.org/articles/67321#video3

constricting the nascent bud, or indirectly with the help of actin polymerization. Interestingly, *Ku et al., 2013* also observed that 60% of assembling particles exhibit a pause around the midway mark of the assembly process. This pause can provide a temporal window for IRSp53 to intervene in the progression of HIV-1 particle assembly as we propose here.

Finally, ESCRT recruitment occurs at the end of virus assembly, after the membrane has been curved, forming a vesicle ready to bud (*Bleck et al., 2014*; *Johnson et al., 2018*). Overexpression of a mutant of the ESCRT protein Tsg101 was previously shown to block HIV-1 budding at a late stage, arresting the budding with a characteristic bulb-shaped phenotype indicative of a defect in the late stage of the bud scission (*Goila-Gaur et al., 2003*), in contrast with our observations with the IRSp53 siRNA phenotype (*Figure 1*, *Figure 1—figure supplement 3*). Consequently, this suggests that Gag-IRSp53 association is ESCRT independent (as shown in *Figure 2—figure supplement 1*) and occurs at an earlier stage of virus assembly.

Another study (*Mercenne et al., 2015*) showed that angiomotin, which acts as an adaptor protein for HIV-1 Gag and the ubiquitin ligase NEDD4L, functions in HIV-1 assembly prior to ESCRT-I recruitment. Interestingly, angiomotin also contains a BAR domain (*Moleirinho et al., 2014*), but it is canonically involved in inducing *positive* curvature, as opposed to the *negative* curvature induced by I-BAR IRSp53. Thus, it is possible that angiomotin functions in another way, for example, at the viral bud neck which has both positive and negative curvatures by facilitating ESCRT recruitment.

IRSp53 itself is a scaffold protein for cofactors of cortical actin signaling (*Zhao et al., 2011*) and we have previously shown that a Rac1 signaling pathway, including IRSp53, is involved in HIV-1 particle production (*Thomas et al., 2015*). Thus, it is possible that IRSp53 could also play a role in generating local cortical actin density in the vicinity of the viral bud in formation. The role of cortical actin associated with IRSp53 scaffolding in that context remains to be elucidated.

Our work illustrates a novel role for the host cellular I-BAR factor IRSp53, which is subverted by the retroviral Gag protein, in HIV-1-induced membrane curvature and in favoring the formation of the fully assembled viral particle.

## Materials and methods

### Antibodies

A rabbit polyclonal anti-IRSp53 antibody (Merck Millipore), a rabbit polyclonal anti-IRTKS (Bethyl), a mouse monoclonal anti-CA (NIH AIDS Reagent Program), a rabbit polyclonal anti-GFP (Invitrogen), a mouse monoclonal anti-human CD63 (Santa Cruz Biotechnology), a mouse monoclonal anti-human CD81 (Santa Cruz Biotechnology), a rabbit monoclonal anti-human TSG101 (Abcam), and a rabbit polyclonal anti-human ALIX (Covalab) and secondary anti-rabbit Atto647N antibody (Sigma) were used in this study.

### Plasmids

The plasmid expressing HIV-1 codon-optimized Gag (pGag(myc), named pGag), the plasmid expressing Pol and Env-deleted HIV-1 (named pNL4.3ΔPolΔEnv was a gift of E.Freed, HIVDRP, NIH, USA) encoding Gag alone with its packageable viral RNA (*Chen et al., 2014b*) and the plasmid expressing full wild-type HIV-1 (named pNL4.3) were described previously (*Favard et al., 2019*). Plasmids IRSp53-GFP, IRTKS-GFP, PinkBAR-GFP, and IRSp53-I-BAR-GFP were obtained from the University of Helsinki (Finland) (*Saarikangas et al., 2009*). Plasmids expressing PH-PLCδ-GFP was a gift of B.Beaumelle (IRIM, France), Gag(i)mCherry (named Gag-mCherry), Gag tagged with

internal photo-activable mEos2 (named Gag(i)mEos2), p6-deleted Gag tagged with mEos2 (named pGagΔp6-mEos2) were described in *Floderer et al., 2018*.

### siRNA

Stealth siRNA (Invitrogen) targeting IRSp53 (BAIAP2) and IRTKS (BAIAP2L2), and Smartpools (Dharmacon) targeting IRSp53 (BAIAP2) or random sequence for siRNA controls were used in this study.

## Cell culture and transfection

Human embryonic kidney cells (HEK 293T-ATCC-CRL-1575TM) were maintained in Dulbecco's Modified Eagle's Medium (DMEM, GIBCO) and human Jurkat T lymphocytes (ATCC-CRL-2899TM) were maintained in RPMI (GIBCO). All cell lines were tested negative for mycoplasma thanks to a MycoAlert Mycoplasma Detection Kit (Lonza Bioscience) done every month. Media was supplemented with 10% fetal bovine serum (FBS, Dominique Dutscher) and complemented with sodium pyruvate and antibiotics (penicillin-streptomycin). Cells were grown at 37°C in a 5% $CO_2$ atmosphere. Transfection was performed by using the calcium phosphate precipitate method on HEK293T cells (as described in *Grigorov et al., 2006*) and the AMAXA (Lonza) method according to the manufacturer's instructions for the Jurkat T cells (as in *Thomas et al., 2015*). Based on different plasmid conditions, the cells were transfected ($2 \times 10^6$ cells/transfection) with a total of 8 µg of plasmids. The amount of transfected plasmid was normalized by adding pcDNA3.1 empty plasmid DNA. The cell medium was replaced by fresh medium 6 hr post-transfection and the experiments were performed 24–48 hr post-transfection. SiRNA transfections in cells were performed with either RNAiMax (Invitrogen) or calcium phosphate buffer in HEK293T cells or by electroporation for Jurkat T cells. One day prior to transfection, $2 \times 10^5$ cells/well were seeded in 2 mL of growth medium without antibiotics in a six-well plate. Transfection was performed using the manufacturer's protocol. After 24 hr of siRNA transfection, the cells were again transfected using the phosphate calcium buffer method. These cells were incubated at 37°C in a 5% $CO_2$ atmosphere for 24/48 hr.

## Immunoprecipitation assay

Based on different plasmid conditions, HEK293T cells ($2 \times 10^6$ cells) were transfected with pGag alone or pGag/pIRSp53-GFP plasmids (8 µg total) and the amount of transfected plasmid was normalized by adding pcDNA3.1 'mock' plasmid. The cell medium was replaced 6 hr post-transfection. After 24 hr post-transfection, the cells were washed with cold $1\times$ phosphate buffer solution (PBS) prior to collection with 800 µL of chilled lysis buffer (50 mM TRIS-HCl [pH = 7.4]; 150 mM NaCl; 1 mM EDTA; 1 mM $CaCl_2$; 1 mM $MgCl_2$; 1% Triton, 0.5% sodium deoxycholate; protease inhibitor cocktail [Roche] one tablet/10 mL lysis buffer). The cells were incubated on ice for 30 min and then centrifuged at 13,000 rpm/15 min/4°C. The supernatant was collected in a new tube and the pellet was discarded. For each condition, 1000 µg of protein (the collected supernatant) was incubated with 1 µg of anti-IRSp53 or anti-GFP antibody on a tube rotator overnight at 4°C. About 25 µL of beads (Dynabeads Protein A, Life Technologies) was added to each tube of protein-antibody complex and incubated for 2 hr on the tube rotator at 4°C. The samples were then washed five times with the lysis buffer, followed by addition of 20 µL 2× Laemmli's buffer to the beads. The samples were denatured at 95°C for 10 min and then processed for Western blot.

## Western blot and analysis

About 50 µg of each protein (intracellular in cell lysates) samples or 20 µL of purified VLP samples were mixed with SDS loading dye, deposited, and resolved on a 10% SDS-PAGE gel. The gels were then transferred on to polyvinylidene PVDF membranes (Amersham). Immunoblotting was performed by incubating the membranes overnight with primary antibody at 4°C, and 2 hr with horseradish-peroxidase (HRP)-conjugated secondary antibody at room temperature. The Western blot signals were detected using ECL Prime/ECL Select substrate (Amersham) and images were taken using ChemiDoc (Bio-Rad).

## VLP purification and quantification

After 24 or 48 hr post-transfection, cell culture supernatants containing Gag-VLPs were collected, filtered through a 0.45 µm filter, and clarified at 800×*g* for 5 min at 4°C. The supernatant was then

purified by loading it on a cushion of 25% sucrose in TNE buffer (25 mM Tris-HCl, 4 mM EDTA, and 150 mM NaCl) and ultracentrifuged at 100,000×$g$ for 1 hr 30 min at 4°C in an SW41Ti rotor (Beckman Coulter). Dry pellets were resuspended in TNE buffer at 4°C overnight. Gag-VLP release was estimated by performing anti-CAp24 immunoblot and by quantifying Gag signal in the blots using Fiji software as described in *Thomas et al., 2015*. The calculation for Gag-VLP release is: % of Gag in VLP = $Gag_{released}/(Gag_{released} + Gag_{intracellular\ normalized\ to\ GAPDH})$.

## Membrane flotation assay

For each condition, $4 \times 10^6$ cells were transfected and viral supernatants were harvested 48 hr post-transfection, as described above. The cells were washed with ice-cold PBS and resuspended in Tris-HCl containing 4 mM EDTA and 1× complete protease inhibitor cocktail (Roche). Every step was then performed at 4°C. Cell suspensions were lysed using a Dounce homogenizer, then centrifuged at 600×$g$ for 3 min to obtain Post-Nuclear Supernatants (PNS). A cushion of 820 µL of 75% (wt/vol) sucrose in TNE buffer was loaded at the bottom of an ultracentrifuge tube and mixed with 180 µL of PNS adjusted to 150 mM NaCl. About 2 mL and 300 µL of 50% (wt/ml) sucrose cushion followed by 0.9 mL of 10% (wt/ml) sucrose cushion were then layered to obtain the gradient that was then centrifuged in a Beckmann SW60Ti rotor at 35,000 rpm, 4°C, overnight. Eight fractions of 500 µL were collected from the top to the bottom and analyzed by Western blotting.

## Transmission electron microscopy

siRNA treated HEK293T cells were fixed in 4% paraformaldehyde and 1% glutaraldehyde in 0.1 M phosphate buffer (pH 7.2) for 48 hr, washed with PBS, post-fixed in 1% osmium tetroxide for 1 hr, and dehydrated in a graded series of ethanol solutions. Cell pellets were embedded in EPON resin (Sigma) that was allowed to polymerize at 60°C for 48 hr. Ultrathin sections were cut, stained with 5% uranyl acetate and 5% lead citrate, and deposited on collodion-coated EM grids for examination using a JEOL 1230 transmission electron microscope.

## Sample preparation for super-resolution PALM/STORM microscopy

HEK293T cells expressing HIV-1 Gag/Gag(i)mEos2 cultured on poly-l-lysine (Sigma) coated 25 mm round #1.5 coverslips (VWR) were fixed using 4% PFA + 4% sucrose in PBS for 15 min at room temperature. Samples were subsequently quenched in 50 mM $NH_4Cl$ for 5 min. Samples were then washed in PBS and then blocked for 15 min iat room temperature using 1% BSA in PBS and subsequently in 0.05% Saponin in 1% BSA in PBS. Samples were stained using a 1:100 dilution of the primary antibodies (rabbit polyclonal anti-human IRSp53, Sigma and rabbit polyclonal anti-human IRTKS antibody, Bethyl) for 60 min at room temperature. Samples were washed three times for 5 min using 1% BSA in PBS followed by 60 min staining using a 1:2000 dilution of the anti-rabbit Atto647N antibody (Sigma). Samples were washed three times for 5 min with PBS and stored in light protected container in +4°C until imaged. Samples were mounted on a StarFrost slide with a silicon joint with the STORM buffer (Abbelight). Cells were imaged within 60 min after application of the STORM buffer.

## PALM/STORM Imaging

Single-molecule localization microscopy was performed on a Nikon inverted microscope equipped with 405, 488, 561, and 642 nm lasers, an EMCCD Evolve 512 Photometrics camera (512 × 512, 16 µm pixel size) with an oil immersion objective 100× NA1.49 Plan Apochromat. PALM imaging of Gag mEos2, activation was performed with laser irradiance set to 0.3 kW/cm$^2$ for 405 nm conversation and ~2.2 kW/cm$^2$ for 561 nm excitation. Illumination was performed over a 25 × 25 µm$^2$ area in the sample (1/e$^2$ spatial irradiance distance) in TIRF-mode. About 20–50,000 images were acquired for each cell with 50 ms integration time. The mean precision localization in PALM measurements was found to be 20 ± 5 nm (*Figure 3—figure supplement 3a*). 2D-STORM imaging of Atto647N was performed using a ~5 kW/cm$^2$ irradiance with the 642 nm excitation. About 25,000 images were acquired for each condition. Tetraspeck 100 nm multicolor beads (Life Technologies) as fiducial markers to correct for drift and chromatic abberation.

## Single-molecule localization microscopy image reconstruction and analysis

SMLM acquisitions were analyzed using the ThunderSTORM plugin in Fiji (*Ovesný et al., 2014*). The mean precision localization in PALM measurements was found to be 20 ± 5 nm (mean ± sd) and 27 ± 9 nm for STORM (*Figure 3—figure supplement 3*). During post-processing, a density filter was applied first to eliminate the background noise by identifying and discarding 'isolated' localizations, with a threshold of a minimum of five neighbors in a 50 nm radius for a molecule to be considered 'not isolated.' In the next step, molecules that converged to the same position were identified as duplicates and removed, keeping the localization with the smallest uncertainty as the valid coordinate. Furthermore, molecules reappearing with one 'off-frame' within 20 nm were merged into one single localization, with the first one appearing considered as the valid localization. Following these steps of post-processing, the sample drift was corrected by the drift correction module using fiducial markers described above. Each acquisition had at least two fiducial markers in the field of illumination. The resulting list of localizations was then used to reconstructing the respective PALM and STORM images and for the CBC analyses using the ThunderSTORM module. The module DBSCAN of the super-resolution quantification software SR Tesseler (*Levet et al., 2015*) was used to analyse the PALM localizations for quantification of Gag cluster sizes. In order to monitor the localization of I-BAR proteins in the vicinity of Gag assembling particles, the Gag particles were segmented by thresholding using Fiji, to generate a binary mask of the PALM images. The centers of each Gag assembling cluster were then determined and a custom MATLAB (Mathworks) code was used to extract localizations in a radius of 80 nm around each Gag cluster center and to extract I-BAR proteins localizations belonging to a disk of 150 nm radius around the center of each Gag clusters (see Results section). These subsets of coordinates were then used to calculate the experimental CBC, with the algorithm developed by *Malkusch et al., 2012* and implemented in the ThunderSTORM plugin of Fiji (see *Figure 3—figure supplement 4*). The CBC values are calculated from single-molecule localization data of two species (Gag and IBAR proteins [IRSp53 or IRTKS]). A CBC value is assigned to each single localization of each species. We analyzed the distributions of these CBC values by plotting and comparing the distribution histograms of the CBCs obtained in the two conditions (IRSp53 vs. IRTKS).

We also performed a set of numerical simulations of super resolution images in Fiji. For this, we generated images by randomly choosing a molecule position within a radius of 60 nm from the cluster center for Gag [a compromise between the values found in the EM images (*Figure 1e*) and those obtained by DBSCAN analysis in the SMLM experimental images (*Figure 3—figure supplement 3*)]. For I-BAR proteins, we randomly assign positions within belts of different waists (from 0 to 100 nm) located at different distances (from 0 to 160 nm) from the Gag cluster center (see *Figure 4* and *Figure 4—figure supplement 1*). Each localization was then randomly assigned a brightness such that the final pointing precision obtained at the end of the process for the Gag localization was equivalent to the experimental one. We then convolved each of these positions by a Gaussian 2D PSF, to generate a diffraction-limited image in which random noise was introduced using the 'add noise' function of Fiji. Super-resolution images of these simulated localizations were then reconstructed using the ThunderSTORM plugin of Fiji with the same parameters as the one used in the experimental data reconstruction. Finally, once the data set of localizations for both numerically simulated Gag clusters and associated surrounding I-BAR proteins were obtained, they were analyzed with the CBC function of ThunderSTORM with the same parameters as in the experimental analysis (analyzing all the neighbor positions within 10 successive radii of 20 nm each to estimate the CBC). Each localization was attributed CBC values and we compared the experimental cumulative distribution functions of CBC values to the simulated cumulative distribution functions using RMSE measurements, in order to establish the most probable configuration of I-BAR belts around the ongoing Gag assembly site.

## Preparation and imaging of fluorescent VLPs

After 24 hr seeding, $2 \times 10^6$ HEK293T cells were transfected with 8 µg of I-BAR-GFP expressing plasmid with or without 8 µg of pGag/pGag(i)mCherry plasmids (2/3 and 1/3, respectively). After 24 hr transfection, cell media (9 mL) was filtered before performing VLPs purification by ultracentrifugation in an SW41Ti rotor (Beckman Coulter) at 29,000 rpm, for 1 hr 30 min, at 4°C, on a

20% sucrose cushion in TNE buffer. Dry pellets were resuspended in 110 µL of TNE and allowed to sediment on round 25 mm coverslips for 45 min in an AttoFluor Cell Chamber (Invitrogen). VLPs were imaged with a Nikon Ti Eclipse 2 TIRF microscope. Images were taken with an Evolve EMCCD camera – 512 photometrics, using a NA = 1.45, 100× objective and using 488 and 561 nm lasers.

### Image analysis for colocalization

Images were acquired with a Zeiss LSM780 (for fixed cells) or a Nikon Eclipse Ti-2 in TIRF mode (for fluorescent viral particles). Colocalization analysis based on Mander's coefficients was performed using JaCOP (Just another Colocalization Plugin) (*Bolte and Cordelières, 2006*). Mander's coefficients are defined as $M1 = \dfrac{\sum_i A_{i,coloc}}{\sum_i A_i}$ and $M2 = \dfrac{\sum_i B_{i,coloc}}{\sum_i B_i}$, $A$ and $B$ being the two respective channels (mCherry and GFP). $0 < M < 1$, with 1 full colocalization and 0.5 random colocalization. The M1 and M2 coefficients were calculated for several images and then represented as column graphs with red columns representing the degree of overlap of mCherry images with GFP images, and green columns representing the inverse.

### Iodixanol gradient

Cell culture medium of HEK293T ($2.5 \times 10^6$ cells plated) transfected with 8 µg of pNL4.3ΔpolΔenv plasmid was collected 48 hr after transfection and filtered using a 0.45 µm filter. The medium was then ultracentrifuged on a 20% sucrose cushion in TNE using an SW41Ti rotor (Beckman) at 40,000 rpm for 1 hr 30 min. A solution with 0.25 M sucrose, 1 mM EDTA, 10 mM Tris-HCl pH 7.4 was used to diluted the 60% iodixanol stock solution (OptiPrep from Sigma) and to prepare a 40% and 20% iodixanol solution. About 1.5 mL of each dilution (60%, 40%, and 20% iodixanol) was successively layered in an SW55Ti tube (Beckman) and the pellet of VLPs obtained after ultracentrifugation on a 20% sucrose cushion in TNE was loaded from the top. Tubes were ultracentrifuged at 50,000 rpm in an SW55Ti rotor (Beckman) at 4°C for 3 hr. Then, 20 fractions of 200 µL were collected from the top of the tube to the bottom. About 20 µL of each fraction was loaded for Western blotting.

### GUV reagents

Brain total lipid extract (131101P) and brain L-α-phosphatidylinositol-4,5-bisphosphate (PIP$_2$, 840046P) were purchased from Avanti Polar Lipids/Interchim. BODIPY-TR-C5-ceramide, (BODIPY TR ceramide, D7540) and Alexa Fluor 488 C5-Maleimide (AX 488) were purchased from Invitrogen. β-casein from bovine milk (>98% pure, C6905) and other reagents were purchased from Sigma-Aldrich. Culture-Inserts 2 Well for self-insertion were purchased from Ibidi (Silicon open chambers, 80209).

### Protein purification and fluorescent labeling

Recombinant mouse IRSp53 I-BAR domain was purified and labeled with AX488 dyes, as previously described (*Prévost et al., 2015*; *Saarikangas et al., 2009*). Recombinant HIV-1 immature Gag protein was purified by J. Mak as described in *Yandrapalli et al., 2016* and labeled with Alexa488 maleimide dyes (Invitrogen). Briefly, a 200 µM solution of the maleimide dye was incubated overnight at 4°C with a 20 µM solution of the Gag purified protein in a buffer of pH 8.0 with 1M NaCl and 50 mM Tris-HCl. Post incubation, the labeled mixture was subjected to dialysis with the Slide-A-Lyzer Mini Dialysis Device (Thermo Fisher Scientific), following the manufacturer's instructions to remove the excess unbound dye from the solution.

### GUV preparation and observation

Lipid and buffer compositions

Lipid compositions for GUVs were brain total lipid extract (*Seong-hyun et al., 2006*) supplemented with 5 mol% brain PI(4,5)P$_2$. If needed, 0.5 mol% BODIPY TR ceramide was present in the lipid mixture as a membrane reporter. The salt buffer inside the GUVs, named I-buffer, was 50 mM NaCl, 20 mM sucrose, and 20 mM Tris-HCl pH 7.5. The salt buffer outside the GUVs, named O-buffer, was 60 mM NaCl and 20 mM Tris-HCl pH 7.5.

## GUV preparation

GUVs were prepared by using the polyvinyl alcohol (PVA) gel-assisted method (*Weinberger et al., 2013*), except GUVs shown in *Figure 6*, *Figure 6—figure supplement 1c* in which the electroformation method was used (*Méléard et al., 2009*). For the PVA method, a PVA solution (5% (w/w) of PVA in a 280 mM sucrose solution) was warmed up to 50°C before spreading on a coverslip that was cleaned in advance by being bath sonicated with 2% Hellmanex for at least 30 min, rinsed with MilliQ water, sonicated with 1 M KOH, and finally sonicated with MilliQ water for 20 min. The PVA-coated coverslip was dried in an oven at 50°C for 30 min. About 5–10 µL of the lipid mixture (1 mg/mL in chloroform) was spread on the PVA-coated coverslip, followed by drying under vacuum for 30 min at room temperature. The PVA-lipid-coated coverslip was then placed in a 10 cm cell culture dish and 0.5 mL of the I-buffer was added on the coverslip, followed by keeping it stable for 45 min at room temperature to allow the GUVs to grow. For the electroformation method, a few µl of lipid mixture at 3 mg/mL were deposited onto platinum electrodes (Goodfellow). The lipid film was dried for at least 30 min under vacuum at room temperature, and then rehydrated in I-buffer under a voltage of 0.25 V and a frequency of 500 Hz overnight at 4°C (*Méléard et al., 2009*).

## Sample preparation and observation

GUVs were first incubated with either Gag or I-BAR domain at bulk concentrations depending on the designed experiments for at least 15 min at room temperature before adding either I-BAR domain or Gag, respectively, into the GUV-protein mixture. In experiments where there was only Gag but no I-BAR domain, the stock solution of I-BAR domain was used in order to obtain a comparable salt strength outside GUVs as those where I-BAR domain was present. The GUV-protein mixture was then incubated for at least 15 min at room temperature before observation. For the Gag/I-BAR membrane recruitment assay, samples were observed on a Nikon C1 confocal microscope equipped with a 60× water immersion objective (Nikon, CFI Plan Apo IR 60× WI ON 1.27 DT 0.17). For the Gag/I-BAR tubulation assay, samples were observed with an inverted spinning disk confocal microscope Nikon eclipse Ti-E, equipped with Yokogawa CSU-X1 confocal head, 100× CFI Plan Apo VC objective (Nikon) and a CMOS camera, Prime 95B (Photometrics).

For all experiments, coverslips were passivated with a β-casein solution at a concentration of 5 g. L$^{-1}$ for at least 5 min at room temperature. Experimental chambers were assembled by placing a silicon open chamber on a coverslip.

## GUV image analysis

Image analysis was performed by using Fiji (*Schindelin et al., 2012*).

### Quantification of AX488 Gag binding on GUV membranes

Fluorescence images were taken at the equatorial planes of GUVs using identical confocal microscopy settings. The background intensity of the AX488 channel was obtained by manually drawing a line with a width of 10 pixels perpendicularly across the membrane of a GUV. We then obtained the background intensity profile of the line where the x-axis of the profile is the length of the line and the y-axis is the averaged pixel intensity along the width of the line. The background intensity was obtained by calculating the mean value of the sum of the first 10 intensity values and the last 10 intensity values of the background intensity profile. To obtain Gag fluorescence intensity on the membrane of the GUV, we used membrane fluorescence signals to find the contour of the GUV (using the 'Fit Circle' function in Fiji). Then, a 10 pixel wide band centered on the contour of the GUV was used to obtain the Gag intensity profile of the band where the x-axis of the profile is the length of the band and the y-axis is the averaged pixel intensity along the width of the band. Gag fluorescence intensity was then obtained by calculating the mean value of the intensity values of the Gag intensity profile, following by subtracting the background intensity.

### Gag sorting map

Fluorescence images of GUVs were taken using identical confocal microscopy settings. For every GUV, we first calculated the fluorescence intensity ratio for every pixel of the Gag and membrane images of a GUV using $\left(I_{Gag} - I_{background}^{Gag}\right)/I_{membrane}$, where $I_{Gag}$ is the Gag intensity, $I_{background}^{Gag}$ is the

background intensity in the Gag channel, and $I_{membrane}$ is the membrane intensity. The sorting map was then obtained by converting the resulting image from the previous step to a pseudo-colored image via the 'Look Up Table, Phase' in Fiji. The background intensity value in the Gag channel was the mean intensity value of a 50 pixel wide square in the background outside GUVs. The sorting map of I-BAR domain was obtained by using the same procedure as those for Gag.

## Statistics

All notched boxes show the median (central line), the 25th and 75th percentiles (the bottom and top edges of the box), the most extreme data points the algorithm considers to be not outliers (the whiskers), and the outliers (crosses).

## Acknowledgements

The authors greatly acknowledge the Montpellier MRI-CNRS and CEMIPAI microscopy facility for access to the PALM/STORM microscopes. The authors thank Eric Freed (NIH, Frederick, MD, USA) for providing the pNL43GagΔPolΔEnv plasmid and A. Cimarelli (CIRI, Lyon, France) for providing the pGag(myc) plasmid. The authors greatly acknowledge the Cell and Tissue Imaging (PICT-IBiSA), Institut Curie, member of the French National Research Infractucture France-BioImaging (ANR10-INBS-04). DM and CF are members of the ImaBio Consortium of the CNRS (GDR ImaBio). This work was supported by the ANRS Grant ECTZ35754. KI was the recipient of an ANRS fellowship for 3 years (2017–2020). RD is a recipient of a SIDACTION fellowship. The authors thank Michael Henderson for careful English reading of the manuscript.

## Additional information

### Competing interests

Patricia Bassereau: Reviewing editor, *eLife*. Pekka Lappalainen: Reviewing editor, *eLife*. The other authors declare that no competing interests exist.

### Funding

| Funder | Grant reference number | Author |
| --- | --- | --- |
| Agence Nationale de Recherches sur le Sida et les Hépatites Virales | ECTZ88374 | Delphine M Muriaux |
| Agence Nationale de la Recherche | ANR10-INBS-04 | Patricia Bassereau |

The funders had no role in study design, data collection and interpretation, or the decision to submit the work for publication.

### Author contributions

Kaushik Inamdar, Data curation, Formal analysis, Validation, Investigation, Visualization, Methodology, Writing - original draft, Writing - review and editing; Feng-Ching Tsai, Data curation, Formal analysis, Validation, Investigation, Methodology, Writing - original draft; Rayane Dibsy, Data curation, Formal analysis, Validation, Methodology; Aurore de Poret, Peggy Merida, Data curation, Formal analysis, Methodology; John Manzi, Resources, Methodology; Remi Muller, Methodology; Pekka Lappalainen, Johnson Mak, Resources, Writing - review and editing; Philippe Roingeard, Data curation, Formal analysis, Supervision, Validation, Visualization, Methodology; Patricia Bassereau, Resources, Supervision, Funding acquisition, Validation, Investigation, Writing - review and editing; Cyril Favard, Data curation, Software, Formal analysis, Supervision, Validation, Investigation, Methodology, Writing - original draft; Delphine Muriaux, Conceptualization, Resources, Data curation, Formal analysis, Supervision, Funding acquisition, Validation, Investigation, Visualization, Methodology, Writing - original draft, Project administration, Writing - review and editing

## Author ORCIDs
Kaushik Inamdar (ID) https://orcid.org/0000-0001-5959-6409
Feng-Ching Tsai (ID) https://orcid.org/0000-0002-6869-5254
Pekka Lappalainen (ID) http://orcid.org/0000-0001-6227-0354
Patricia Bassereau (ID) http://orcid.org/0000-0002-8544-6778
Cyril Favard (ID) https://orcid.org/0000-0002-8304-2980
Delphine Muriaux (ID) https://orcid.org/0000-0001-8517-9342

## Decision letter and Author response
Decision letter https://doi.org/10.7554/eLife.67321.sa1
Author response https://doi.org/10.7554/eLife.67321.sa2

## Additional files

### Supplementary files
- Supplementary file 1. Key Resources Table 1.
- Transparent reporting form

### Data availability
All data have been provided in the manuscript and supporting files in our submission that allows research reproductibility (see source data, reagents table and supplemental informations).

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
