## [Decision Letter]

**Acceptance summary:**

This manuscript combines cell biology, biochemistry, and quantitative biophysics to understand how a new host cell factor, the human I-BAR domain protein IRSp53, promotes HIV type 1 (HIV-1) assembly and release. Since this new factor is a protein involved in the generation and sensing of negative membrane curvature, this manuscript will be of interest not only for retrovirologists and virologists in general but also for membrane biologists and biophysicists.

**Decision letter after peer review:**

Thank you for submitting your article "Full assembly of HIV-1 particles requires assistance of the membrane curvature factor IRSp53" for consideration by *eLife*. Your article has been reviewed by 3 peer reviewers, including Felix Campelo as the Reviewing Editor and Reviewer #1, and the evaluation has been overseen by Vivek Malhotra as the Senior Editor. The following individuals involved in review of your submission have agreed to reveal their identity: Ricardo Henriques (Reviewer #2); Zandrea Ambrose (Reviewer #3).

Essential revisions:

We all agreed that this paper provides novel and important new data that will merit, after revision, publication in *eLife*. We believe that for the revision, we are not asking the authors for extensive extra work, just some key experimental controls or tests that could be duly due in a few months. We'd also appreciate that the authors go through a thorough revision of the text, in terms of clarity and quality (also on the figures and method description, see details below).

To summarize, the main points that the reviewers considered necessary for this article to be accepted at *eLife* are:

1) Improve quality of the IP/co-IP data as well as the biochemical measurements of membrane fraction pools.

2) Assess specificity of siRNA-mediated knockdowns.

3) It is unclear what the role of IRSp53 is in the membrane curvature of CD4^+^ T cells and whether expression levels and localization are consistent with Jurkat T cells.

4) Better description and quantitative comparison of simulated and experimental SMLM data.

5) Make sure that method descriptions are complete.

6) Discuss the very high similarity between histograms in Figure 3c,d (HEK293T) and Figure S10c,d (Jurkats).

Since the reviews are quite detailed and include very valuable comments that will for sure improve the quality of the manuscript, we include them in detail in this decision letter. Please, consider them as much as possible when submitting a revised version.

*Reviewer #1 (Recommendations for the authors):*

– Figure 1C: How specific is this? Can the authors express siRNA-resistant mutants of IRSp53 and see if this reverts the phenotype? Or at least test the effect with other siRNA oligoes?

– Line 152: To overcome the limitation caused by the IgGs masking the IRSp53 signal in the WBs, there are nowadays different ways of probing the membranes with secondary antibodies that do not detected denatured IgGs (available commercially from different companies). Along these lines, is this Ig G band the band seen in Figure S4a between the 55 and 70 kDa marker bands? What is the mature MW of IRSp53? Is it 53 kDa or higher? If 53 kDa, then IRSp53-GFP should be around 80 kDa (53+27) and not >100 kDa as in the IP in S4A. Could the authors discuss about this?

– Overexpression of IRSp53-GFP does not lead to more Gag being pulled down (lanes 3 and 4 in S4a). Is the GFP-tagged form of IRSp53 functional (at least in respect to Gag binding)? Could the authors elaborate a bit more on this?

– Figure 2c: To asses if the increased binding of IRSp53 to membranes in the presence of Gag is specific for this IBAR protein, a control where the membrane bound pool of another IBAR protein (e.g. IRTKS) should be included.

– This is possible obvious for people in the field, but I missed at understanding why relatively low concentration (0.05 uM) of IRSp53 IBAR induces invagination into GUVs but 10x larger concentration does not seem to deform those GUVs (Figure 5). Could the authors clarify this please?

*Reviewer #2 (Recommendations for the authors):*

1. Gag and IRSp53 complexing is shown in an immunoprecipitation assay (Figure 2a). However, the quantification of membrane fraction enrichment of IRSp53 upon Gag expression in the membrane floatation assay the presented data is not convincing. Inamdar and colleagues claim a 2-fold increase of IRSp53 in the membrane fraction. Still, based on the dataset shown in Figure 2c, the findings don't seem to match the minor differences observed in the respective bands in the gel image. In the presented data set, the membrane fractions of Gag and IRSp53 do not appear in the same lane (Lane 1). I, therefore, suggest that the authors provide a more detailed description of their analysis method and the full data set to convincingly confirm the difference in membrane-bound IRSp53 upon Gag-expression compared to control cells, and the amount of membrane-bound Gag relative to its cytosolic counterpart. If the previously reported results hold, cross-validate them using a different methodology, for instance, by measuring the colocalisation of IRSp53 and Gag with the plasma membrane using microscopy. If the new calculations do not match the results previously reported, the authors should adjust their conclusions accordingly.

2. Dual-color SMLM is used to elucidate the IPSp53/IRTKS organisation patterns at Gag clusters sites by PALM/dSTORM imaging. Inamdar and colleagues quantify the Gag cluster size distribution and show that IPSp53 shows a considerable colocalized fraction, while IRTKS tends toward an anticorrelated distribution. In the next step, the authors try to deduce the spatial organisation of IPSp53 and IRTKS around Gag clusters based on simulated datasets of Gag and IPSp53/IRTKS CBC distribution geometries.

While dual-color SMLM is the right approach, the study presented here will benefit from a consistent and comprehensive description of the experimental details (see also Mn7) and the simulation. I propose adding a schematic of the geometric simulation parameters shown in Figure S8a in Figure 3. Also, it should be highlighted in Figure 3e and 3f that the simulated distributions presented are based on different waist diameters.

A second point that should be addressed is the lack of a quantitative measure of how well the experimental results are described by the simulated data. Also, the authors should comment on the effects of the choice of the radius in the data filtering (as shown in Figure S7) and over parameterization.

3. In the controlled in-vitro environment of GUVs, the authors show that Gag membrane recruitment is enhanced in the presence of the IRSp53 I-Bar domain and preferentially locates to the areas of high curvature in I-Bar domain induced membrane tubes. While I fully support the conclusions drawn from this elegant experiment, a complete description of the GUV preparation and imaging methods is missing. Also, from the methods section and the captions in Figure 5 and S11, it is unclear which membrane label was used in the experiment.

*Reviewer #3 (Recommendations for the authors):*

It would have been better to incorporate Jurkat CD4^+^ T cell data into the main figures to highlight the biological relevance of the findings.

Additional recommendations to strengthen the manuscript:

1) The role and expression levels of IRSp53 in CD4^+^ T cells should be described and/or shown, as these are the relevant target cells for HIV-1 infection.

2) It is recommended that "gene extinction" in lines 105 and 107 be revised to "protein depletion" or "reduced gene expression" or something similar, as siRNA do not affect the gene but rather gene expression.

3) Figure 1e (both schematic and graph data points) is nearly impossible to see as it is small and very faint. It is recommended that darker colors and thicker lines and symbols be used, similar to other figures in the manuscript.

4) The order of the supplemental figures should be in chronological order from when they are referenced in the text, which is not the case in this manuscript. And all parts of each figure should be described in the text (e.g. Figure S1a).

5) It is suggested that the reference of GUVs on page 7 be removed. Otherwise, a description of what they are and the significance of the finding described here should be included.

6) Likewise, description of Gag(i)mEos2 is referred on page 8 but it is not clear why this construct was used. Presumably this was to show that the Gag used in the imaging studies behaved similarly to wild-type Gag, but this is not clear.

7) On lines 192-4, it is stated, "We evidenced that cellular Gag expression, most probably by triggering Rac1 activation (Thomas et al., 2015), favors cell membrane binding of IRSp53." As Rac1 was not evaluated here, it is suggested that the sentence be modified to, "We observed that cellular Gag expression, possibly by triggering Rac1 activation (Thomas et al., 2015), favors cell membrane binding of IRSp53."

8) The manuscript overall would benefit from strong editing for grammar, punctuation, and better word usage.

---

## [Author Response]

Essential revisions:We all agreed that this paper provides novel and important new data that will merit, after revision, publication in eLife. We believe that for the revision, we are not asking the authors for extensive extra work, just some key experimental controls or tests that could be duly due in a few months. We'd also appreciate that the authors go through a thorough revision of the text, in terms of clarity and quality (also on the figures and method description, see details below).To summarize, the main points that the reviewers considered necessary for this article to be accepted at eLife are:1) Improve quality of the IP/co-IP data as well as the biochemical measurements of membrane fraction pools.

The IP/co-IP datas have been update with new experiments that summarize all the points of the reviewers, including new controls (Input, IP/coIP, and FlowThrough), better quality gels, and better explanation in Materials and methods section for sake of clarity. See new figure 2b.

Biochemical measurements of membrane fraction pools were made several times (and with 3 independent people in the lab – KI, RD and PM), so we have now included better western blot images of membrane flotation assays that fit with the graph (see figures 2c,d); we also added the ribosomal S6 marker as a cytosolic control fraction, and IRTKS western blotting of the fractions.

2) Assess specificity of siRNA-mediated knockdowns.

We have tried several different commercial and home-designed siRNA targeting IRSp53 from different companies (providing single siRNA and multiple siRNA mix): we have summarizing all in Author response image 1. One can see that indeed only 2 siRNA were effective in extinguishing IRSp53 gene: one from Invitrogen on endogenous IRSp53 and ectopic IRSp53-GFP and one from Dharmacon that was only effective on ectopic IRSp53-GFP, as revealed by Western Blot (Author response image 1). Furthermore, the specificity of the siRNA was challenge by testing siRNA IRSp53 on human IRSp53-GFP and on mouse I-BAR-GFP in HEK293T transfected cells and visualized by fluorescence microscopy. Results show in Author response image 1 that only siIRSp53 is able to extinguished human IRSp53-GFP and not mouse I-BAR-GFP. SiIRTKS and siCtrl are not extinguishing any of these genes. Overall these results confirm the specificity of IRSp53 siRNA-mediated knockdowns.

**Author response image 1. respfig1:** Specificity of siRNA-mediated knockdowns. (**A**) Western blots of HEK293T cells lysates probed with anti-IRSp53 antibody (and house-keeping gene GAPDH) showing a series of different siRNA IRSp53 (and siRNA Control, CTRL from Invitrogen, Dharmacon or Σ) on endogenous and ectopic IRp53 genes in human HEK293T cells and their efficacy in specifically down regulating IRSp53. (**B**) siRNA IRSp53 from Invitrogen was tested for its specificity in extinguishing human IRSp53-GFP protein expressed in transfected HEK293T cells, but not mouse I-BAR-GFP, and as compare to siRNA control and IRTKS, revealed by fluorescence imaging (GFP).

To further answer the reviewers’ comments, we also perform one rescue experiment of the phenotype as shown Author response image 2. We observed that, upon co-transfection of pGag+pIRSp53-GFP+siRNA IRSp53 (lane 2), about 50% of the ectopic IRSp53-GFP was extinguished (since this construct is not siRNA resistant), leaving 50% of this ectopic protein expressed in the cells. In this context, one can observe that Gag-VLP release is ~50% (lane 2), similar to the condition pGag+siCTRL (lane 3). When we compare this to pGag+siIRSp53 (lane 4) which is reduced by 2-3 fold (data from Figure 1b of the manuscript), we can say that the remaining IRSp53-GFP in the Lane 2 seems to rescue the defect caused by extinction of the endogenous IRSp53. In the condition pGag+pIRSp53-GFP +siCTRL, VLP-Gag release was slightly reduced. This is an atypical rescue experiment since we do not have an IRSp53-GFP that is resistant to the siRNA IRSp53 used in this study (Author response image 1), but it suggests that if IRSp53-GFP is overexpressed in the presence of Gag and the siRNA IRSp53, VLP-Gag release is at a normal 50% level in contrast to the absence of IRSp53-GFP (compare lane 2 with lane 4). Unfortunately, due to limited time and by the siRNA IRSp53 out of stock, and the delay in supply, we could only provide one experiment. We thus decided to show it for answering the reviewers but not as part of a figure in the final manuscript.

**Author response image 2. respfig2:** Rescue of siRNA IRSp53 knock-down with overexpression of IRSp53-GFP. 293T cell were transfected with pGag, pIRSp53 and siRNA control (siCTRL, lane 1) or siRNA IRSp53 (lane 2); cell lysat and VLP wre loaded on SDS-PAGE gels and immunoblots were revealed with anti-GFP (for IRSp53-GFP) and anti-CAp24 (for HIV-1 Gag). One graph on the left shows the percentage of IRSp53-GFP expression upon siRNA IRSp53 cell treatment (lane 2) as compare to the siRNA CTRL (lane 1). The graph on the right shows the resulting gel quantification for the % of Gag-VLP release upon siRNA IRSp53 cell treatment (lane 2) as compare to the siRNA CTRL (lane 1) in the presence of IRSp53-GFP over-expression, or without (lane 3 and 4, as in Figure 1b). N=1 rescue experiment.

3) It is unclear what the role of IRSp53 is in the membrane curvature of CD4^+^ T cells and whether expression levels and localization are consistent with Jurkat T cells.

We have published that IRSp53 (using siRNA) is involved in HIV-1 particle release on primary T cells (PBMC derived T cells) in Thomas et al., JVI 2015, so high probability is that it would be the same in different cell type, transfected HEK293T cells, transfected or infected Jurkat T cells and infected primary T cells. But we have not done the extensive super-resolution microscopy on infected primary T cells because this would require time overconsuming study. We are currently proceeding in setting up condition with an infectious HIV-1 virus carrying mEOS2 photoactivable protein for being able to infect primary T cells and go on for further research using infectious relevant system and super-resolution microscopy, but it is not ready for this current manuscript as it would require months of extra work and experiments.

Although, we agree with the reviewer #3 that the localization of Gag in Jurkat T cells and in primary CD4 T cells is different at the cellular level, but at the nanoscopic level of the budding sites, chances are that it would be similar (to be checked in further studies).

4) Better description and quantitative comparison of simulated and experimental SMLM data.

We have addressed this point in the manuscript by

i) describing in detail the method used to record, extract and analyse the experimental data, for this we have added extended description of methods used in quantifying SMLM data, we have introduced a section in the methods explaining how we performed the simulations. We have also described more in details the analysis of SMLM data in the Results section.

ii) performing root mean square error quantification in order to compare the experimental to the simulated cumulative distribution functions. We have added maps of values of the RMSE to show similarities between the different numerical schemes (waists and distance from the center of the surrounding belt) and the experimental data in the supplemental figure 9 (FigS9). We have changed the text of the manuscript in the Results section accordingly. We want to point out here that the idea of these simulations is not to find a perfect similarity between experiments and simulation distribution of the CBC value. This numerical simulation has been performed more in a view of discriminating among different possible scenarii of I-BAR proteins (IRSp53 or IRTKS) localisation around a Gag on going bud. We show here the scenario that fits the best to the experimental CBC distribution in order to illustrate that there are clear differences in between IRSp53 and IRTKS in terms of organisation around the on-going bud.

5) Make sure that method descriptions are complete.

We improved the method descriptions, by extending it and going more into details, in particular for the SMLM and simulation analysis but also for the transfection methods and the GUV data analysis.

6) Discuss the very high similarity between histograms in Figure 3c,d (HEK293T) and Figure S10c,d (Jurkats).

The very high similarity between histograms of cbc values in the case of HEK 293T and Jurkats T cells suggest that the organisation of these two I-Bar proteins around the budding site is independent of the cell type.

We did not discuss it in the previous version of the manuscript as we found this result to be not so surprising. In fact, we and others have previously shown that, on average, the bud assembly kinetics was independent of the cell type (Floderer et al., Sci Reports 2018; Ivachenko et al., Plos Pathogens 2009). Interestingly, we also observe (as others) that the final diameters of the released particles are equivalent independently of the cell type where they are produced, Jurkat T cells (Floderer et al., Sci Reports 2018); HEK 293T (this study), HeLa cells (Manley et al., 2009). These two mechanics and kinetics features are therefore very similar independently of the cell type, suggesting that the time evolution of the curvatures generated during the assembly of the virus is certainly equivalent in both cell type. Supposing that these processes rely on the same molecular interplay with the same kinetics, with Gag as an orchestrator and knowing that IRSp53 as well as IRTKS proteins are express in both cell types (Jurkat and HEK 293T) where they recognise equivalent curvatures and have the same functions at the plasma membrane, we think that these observations can explain why this colocalization based CBC histograms are very similar.

Since the reviews are quite detailed and include very valuable comments that will for sure improve the quality of the manuscript, we include them in detail in this decision letter. Please, consider them as much as possible when submitting a revised version.Reviewer #1 (Recommendations for the authors):– Figure 1C: How specific is this? Can the authors express siRNA-resistant mutants of IRSp53 and see if this reverts the phenotype? Or at least test the effect with other siRNA oligoes?

See the above general response to the reviewers. All the answers to these questions are presented in Author response images 1 and 2 to confirm siRNA IRSp53 specificity.

– Line 152: To overcome the limitation caused by the IgGs masking the IRSp53 signal in the WBs, there are nowadays different ways of probing the membranes with secondary antibodies that do not detected denatured IgGs (available commercially from different companies). Along these lines, is this Ig G band the band seen in Figure S4a between the 55 and 70 kDa marker bands? What is the mature MW of IRSp53? Is it 53 kDa or higher? If 53 kDa, then IRSp53-GFP should be around 80 kDa (53+27) and not >100 kDa as in the IP in S4A. Could the authors discuss about this?– Overexpression of IRSp53-GFP does not lead to more Gag being pulled down (lanes 3 and 4 in S4a). Is the GFP-tagged form of IRSp53 functional (at least in respect to Gag binding)? Could the authors elaborate a bit more on this?

In order to simplify and overcome the difficulty to read IP/co-IP with IRSp53 (53 KDa) and Gag (55KDa) due to IgG similar sizes, we perform new IP/co-IP experiment with ectopic IRSp53-GFP (80KDa); the results are shown in the new figure 2b that present very clear results on IRSp53-GFP/Gag complexing. Thus, the figure S4a has been removed.

As for the “real” migration size of IRSp53 (below 55KDa) and IRSp53-GFP (below 100KDa, between 70 and 100 KDa) on SDS-PAGE gels, we cannot explain better, but we observe using the siRNA IRSp53 (see Figure 1 – Supplemental 1, Author response images 1 and 2) that we are indeed knocking down IRSp53 or ectopic IRSp53-GFP. The other band above MW 55KDa is indeed IRSp58 another isoform of IRSp53/58 that has been reported [for example, see DOI: 10.1523/JNEUROSCI.19-17-07300. 1999], and that can be seen in the IRSp53 immunoblot in figure S1d.

– Figure 2c: To asses if the increased binding of IRSp53 to membranes in the presence of Gag is specific for this IBAR protein, a control where the membrane bound pool of another IBAR protein (e.g. IRTKS) should be included.

As suggested by the reviewer, IRTKS has been included in the new figure 2d. The gel band quantification reveals that IRTKS membrane binding is not changing upon Gag expression, contrary to IRSp53 or Tsg101.

– This is possible obvious for people in the field, but I missed at understanding why relatively low concentration (0.05 uM) of IRSp53 IBAR induces invagination into GUVs but 10x larger concentration does not seem to deform those GUVs (Figure 5). Could the authors clarify this please?

The reviewer is right about this seemingly nontrivial observation that puzzles us as well. We verified this point by measuring the percentages of GUVs having inward membrane tubes generated by the I-BAR domain at different bulk concentrations outside the GUVs. We observed an increase of GUVs having tubes at concentrations ranging from 0.005 μM to 0.06 μM; however, when we increased I-BAR concentrations from 0.1 μM up to 1 μM, we observed a decrease in the amount of GUVs having tubes. See the following table for the quantification.

**Author response table 1. resptable1:** 

I-BAR concentration	Percentage of GUVs with tubes (%)	Total number of GUVs
0.005 uM	24	78
0.02 uM	90	260
0.06 uM	87	189
0.1 uM	22	159
0.2 uM	22	120
0.5 uM	19	123
1 uM	8	77

Given that the observed concentration-dependent tubulation of I-BAR domain is not the focus of this paper, we decided not to include the abovementioned results. However, to clarify the point mentioned by the reviewer, we have added the following sentences in the manuscript:

“Previous in vitro studies showed that when placing IRSp53 I-BAR domain outside PIP2-containing GUVs, I-BAR domain can deform GUV membranes, generating tubes towards the interior of the vesicles (Jarin et al., 2019; Prévost et al., 2015; Saarikangas et al., 2009). […] Future work is required to investigate on this seemingly puzzled observation.”

If the reviewer thinks that these results would be useful for the readers, we could include the abovementioned quantification in SI.

Reviewer #2 (Recommendations for the authors):1. Gag and IRSp53 complexing is shown in an immunoprecipitation assay (Figure 2a). However, the quantification of membrane fraction enrichment of IRSp53 upon Gag expression in the membrane floatation assay the presented data is not convincing. Inamdar and colleagues claim a 2-fold increase of IRSp53 in the membrane fraction. Still, based on the dataset shown in Figure 2c, the findings don't seem to match the minor differences observed in the respective bands in the gel image. In the presented data set, the membrane fractions of Gag and IRSp53 do not appear in the same lane (Lane 1). I, therefore, suggest that the authors provide a more detailed description of their analysis method and the full data set to convincingly confirm the difference in membrane-bound IRSp53 upon Gag-expression compared to control cells, and the amount of membrane-bound Gag relative to its cytosolic counterpart. If the previously reported results hold, cross-validate them using a different methodology, for instance, by measuring the colocalisation of IRSp53 and Gag with the plasma membrane using microscopy. If the new calculations do not match the results previously reported, the authors should adjust their conclusions accordingly.2. Dual-color SMLM is used to elucidate the IPSp53/IRTKS organisation patterns at Gag clusters sites by PALM/dSTORM imaging. Inamdar and colleagues quantify the Gag cluster size distribution and show that IPSp53 shows a considerable colocalized fraction, while IRTKS tends toward an anticorrelated distribution. In the next step, the authors try to deduce the spatial organisation of IPSp53 and IRTKS around Gag clusters based on simulated datasets of Gag and IPSp53/IRTKS CBC distribution geometries.While dual-color SMLM is the right approach, the study presented here will benefit from a consistent and comprehensive description of the experimental details (see also Mn7) and the simulation. I propose adding a schematic of the geometric simulation parameters shown in Figure S8a in Figure 3. Also, it should be highlighted in Figure 3e and 3f that the simulated distributions presented are based on different waist diameters.

We thank the reviewer for his suggested improvements leading to a better comprehension of our work by the future reader. We now have changed the figures and split the experimental part and the simulated part of the dual-colour SMLM into 2 different figures, namely figure 3 and figure 4. We have added the geometric simulation parameters in figure 4 and highlighted that the simulated distributions are based on different waist of the assembly surrounding belt.

A second point that should be addressed is the lack of a quantitative measure of how well the experimental results are described by the simulated data.

In order to quantify the similarity between experimental and simulated CBC, we performed a RMSE quantification of the difference between experimental and simulated CBC cumulative distribution functions as these functions are the one represented in figure 4. A map of the different RMSE values obtained for all the different parameters used in the simulation is depicted in the supplementary figure (Figure 4 – supplemental 1). The minimal RMSE in the maps is found for a distance of 80nm and a belt waist of 40 nm in the case of IRSp53, whereas this minimum is displaced towards higher distance (140 nm) and larger waist (100 nm) in the case of IRTKS. However, the idea of performing simulation with this simple configurations (belts of IRSp53/IRTKS surrounding Gag assembly) was not to retrieve the experimental configuration observed here but to (i) show that differences observed here in CBC can be interpreted as different geometries of I-BAR proteins (Irsp53/IRTKS) surrounding HIV-Gag on going assemblies and (ii) relate that on average, surrounding IRSp53 belt is smaller and located closer to the centre of the assembly than IRTKS is.

Also, the authors should comment on the effects of the choice of the radius in the data filtering (as shown in Figure S7) and over parameterization.

We have added different sentences to precise these choices (data filtering and parametrization) in the main manuscript, as well as in the supplementary figures.

Regarding the choice of the radius for the data filtering, we have justified it in the results (line 215-221):

“We then kept all HIV-1 Gag located within a distance of 80 nm from this center (70 to 80% of the all clusters size described above are found within this distance) and all the IRSp53 (or IRTKS) found in a distance of 150 nm from this center (~2x the assembly site size, see Figure 3 – supplemental 4 for details on the process workflow). We choose this IRSp53 (or IRTKS) cut-off distance to avoid cross-colocalization between different HIV-1 Gag clusters in dense areas.”

Regarding the parameterization, we have introduced the following text in the Results section (line 240-248):

“Although CBC values gives a quantitative value of the colocalization, it does not provide direct information on the average positions of IRTKS or IRSp53 molecules with respect to Gag molecules within the assembling clusters were unclear. […] Thus, to gain more insight into these colocalisation quantification, we performed simulations to generate different patterns of PALM/STORM localizations (Figure 4), and analyzed them with the same set of parameter (total distance and number of circles) that the one we used for the experimental data.”

We hope these comments and justification answer to the referee’s suggestion and somehow maybe make our final statements more precisely related to the method we used.

3. In the controlled in-vitro environment of GUVs, the authors show that Gag membrane recruitment is enhanced in the presence of the IRSp53 I-Bar domain and preferentially locates to the areas of high curvature in I-Bar domain induced membrane tubes. While I fully support the conclusions drawn from this elegant experiment, a complete description of the GUV preparation and imaging methods is missing. Also, from the methods section and the captions in Figure 5 and S11, it is unclear which membrane label was used in the experiment.

We thank the reviewer for pointing this out. In Figure 5 (now Figure 6) and Figure 6—figure supplement 2 (now Figure 6—figure supplement 1), 0.5mole% of BODIPY-TR-C5-ceramide is incorporated in GUV membranes for visualizing the membranes. We have clarified this in the figure legends. We have also gone through the GUV preparation and imaging method descriptions to make sure they are completed.

Reviewer #3 (Recommendations for the authors):It would have been better to incorporate Jurkat CD4^+^ T cell data into the main figures to highlight the biological relevance of the findings.Additional recommendations to strengthen the manuscript:1) The role and expression levels of IRSp53 in CD4^+^ T cells should be described and/or shown, as these are the relevant target cells for HIV-1 infection.

Currently, the very few results that we have on IRSp53 in HIV infected primary CD4 T cells are too preliminary and cannot be reasonably included in this study.

2) It is recommended that "gene extinction" in lines 105 and 107 be revised to "protein depletion" or "reduced gene expression" or something similar, as siRNA do not affect the gene but rather gene expression.

Thank you for pointing that out. We have correcting it all over the main text by saying “siRNA knock-down gene expression”.

3) Figure 1e (both schematic and graph data points) is nearly impossible to see as it is small and very faint. It is recommended that darker colors and thicker lines and symbols be used, similar to other figures in the manuscript.

The reviewer is right. Thus, all the letters/writing in the figure 1 have been increased in size in agreement with the reviewer’s comment, which makes the new figure 1 more readable.

4) The order of the supplemental figures should be in chronological order from when they are referenced in the text, which is not the case in this manuscript. And all parts of each figure should be described in the text (e.g. Figure S1a).

We apologize for this error that is now corrected in the new version of the manuscript.

5) It is suggested that the reference of GUVs on page 7 be removed. Otherwise, a description of what they are and the significance of the finding described here should be included.

This sentence was removed from the part 2 of the results (page 7). In the current manuscript, we described this GUV experiment and results in section 5 together with all the GUV experiments. Then the description of Figure 6—figure supplement 1a, b results was inserted in the section 5 (Page 14, lanes 334-339) and the significance of the finding is now included in the results and in the Discussion section (page 15 lanes 365-66).

6) Likewise, description of Gag(i)mEos2 is referred on page 8 but it is not clear why this construct was used. Presumably this was to show that the Gag used in the imaging studies behaved similarly to wild-type Gag, but this is not clear.

As it is written page 8, Gag and Gag(i)mEOS2 were both pull down with IRSp53 during IP/co-IP. We had to check that Gag(i)mEOS2 that we used for PALM/STORM microscopy imaging was immunoprecipitated the same as Gag. So, yes, it was to show that the labelled Gag(i)mEOS2 protein used in the imaging studies behaved similarly to wild-type Gag. For a sake of clarity, we added one sentence lane 188 page 8.

7) On lines 192-4, it is stated, "We evidenced that cellular Gag expression, most probably by triggering Rac1 activation (Thomas et al., 2015), favors cell membrane binding of IRSp53." As Rac1 was not evaluated here, it is suggested that the sentence be modified to, "We observed that cellular Gag expression, possibly by triggering Rac1 activation (Thomas et al., 2015), favors cell membrane binding of IRSp53."

Thank you; this has been modified accordingly to the reviewer’s advice (page 8, lanes 193-194).

8) The manuscript overall would benefit from strong editing for grammar, punctuation, and better word usage.

The manuscript in its final new version has been given to a native English for editing.